# Breaking the Communication-Privacy-Accuracy Tradeoff with $f$-Differential Privacy

**Richeng Jin[1]**   **Zhonggen Su[1]**   **Caijun Zhong[1]**
**Zhaoyang Zhang[1]**   **Tony Q.S. Quek[2]**   **Huaiyu Dai[3]**

[1]Zhejiang University   [2]Singapore University of Technology and Design
[3]North Carolina State University

## Abstract

We consider a federated data analytics problem in which a server coordinates the collaborative data analysis of multiple users with privacy concerns and limited communication capability. The commonly adopted compression schemes introduce information loss into local data while improving communication efficiency, and it remains an open problem whether such discrete-valued mechanisms provide any privacy protection. In this paper, we study the local differential privacy guarantees of discrete-valued mechanisms with finite output space through the lens of $f$-differential privacy (DP). More specifically, we advance the existing literature by deriving tight $f$-DP guarantees for a variety of discrete-valued mechanisms, including the binomial noise and the binomial mechanisms that are proposed for privacy preservation, and the sign-based methods that are proposed for data compression, in closed-form expressions. We further investigate the amplification in privacy by sparsification and propose a ternary stochastic compressor. By leveraging compression for privacy amplification, we improve the existing methods by removing the dependency of accuracy (in terms of mean square error) on communication cost in the popular use case of distributed mean estimation, therefore breaking the three-way tradeoff between privacy, communication, and accuracy.

## 1 Introduction

Nowadays, the massive data generated and collected for analysis, and consequently the prohibitive communication overhead for data transmission, are overwhelming the centralized data analytics paradigm. Federated data analytics is, therefore, proposed as a new distributed computing paradigm that enables data analysis while keeping the raw data locally on the user devices [1]. Similarly to its most notable use case, i.e., federated learning (FL) [2, 3], federated data analytics faces two critical challenges: data privacy and communication efficiency. On one hand, the local data of users may contain sensitive information, and privacy-preserving mechanisms are needed. On the other hand, the user devices are usually equipped with limited communication capabilities, and compression mechanisms are often adopted to improve communication efficiency.

Differential privacy (DP) has become the gold standard for privacy measures due to its rigorous foundation and simple implementation. One classic technique to ensure DP is adding Gaussian or Laplacian noises to the data [4]. However, they are prone to numerical errors on finite-precision computers [5] and may not be suitable for federated data analytics with communication constraints due to their continuous nature. With such consideration, various discrete noises with privacy guarantees have been proposed, e.g., the binomial noise [6], the discrete Gaussian mechanism [7], and the Skellam mechanism [8]. Nonetheless, the additive noises in [7] and [8] assume infinite range, which renders them less communication-efficient without appropriate clipping. Unfortunately, clipping

37th Conference on Neural Information Processing Systems (NeurIPS 2023).

usually ruins the unbiasedness of the mechanism. [9] develops a Poisson binomial mechanism (PBM) that does not rely on additive noise. In PBM, each user adopts a binomial mechanism, which takes a continuous input and encodes it into the success probability of a binomial distribution. The output of the binomial mechanism is shared with a central server which releases the aggregated result that follows the Poisson binomial distribution. However, [9] focuses on distributed DP in which the server only observes the output of the aggregated results instead of the data shared by each individual user, and therefore, requires a secure computation function (e.g., secure aggregation [3]).

In addition to discrete DP mechanisms, existing works have investigated the fundamental tradeoff between communication, privacy, and accuracy under the classic $(\epsilon, \delta)$-DP framework (e.g., [10, 11, 12, 13]). Notably, in the case of distributed mean estimation, [13] incorporates Kashin's representation and proposed Subsampled and Quantized Kashin's Response (SQKR), which achieves order-optimal mean square error (MSE) that has a linear dependency on the dimension of the private data $d$. SQKR first computes Kashin's representation of the private data and quantizes each coordinate into a 1-bit message. Then, $k$ coordinates are randomly sampled and privatized by the $2^k$-Random Response mechanism [14]. SQKR achieves an order-optimal three-way tradeoff between privacy, accuracy, and communication. Nonetheless, it does not account for the privacy introduced during sparsification.

Intuitively, as compression becomes more aggressive, less information will be shared by the users, which naturally leads to better privacy protection. However, formally quantifying the privacy guarantees of compression mechanisms remains an open problem. In this work, we close the gap by investigating the local DP guarantees of discrete-valued mechanisms, based on which a ternary stochastic compressor is proposed to leverage the privacy amplification by compression and advance the literature by achieving a better communication-privacy-accuracy tradeoff. More specifically, we focus on the emerging concept of $f$-DP [15] that can be readily converted to $(\epsilon, \delta)$-DP and Rényi differential privacy [16] in a lossless way while enjoying better composition property [17].

**Our contributions**. In this work, we derive the closed-form expressions of the tradeoff function between type I and type II error rates in the hypothesis testing problem for a generic discrete-valued mechanism with a finite output space, based on which $f$-DP guarantees of the binomial noise (c.f. Section 4.1) and the binomial mechanism (c.f. Section 4.2) that covers a variety of discrete differentially private mechanisms and compression mechanisms as special cases are obtained. Our analyses lead to tighter privacy guarantees for binomial noise than [6] and extend the results for the binomial mechanism in [9] to local DP. To the best of our knowledge, this is the first work that investigates the $f$-DP guarantees of discrete-valued mechanisms, and the results could possibly inspire the design of better differentially private compression mechanisms.

Inspired by the analytical results, we also leverage the privacy amplification of the sparsification scheme and propose a ternary stochastic compressor (c.f. Section 5). By accounting for the privacy amplification of compression, our analyses reveal that given a privacy budget $\mu$-GDP (which is a special case of $f$-DP) with $\mu < \sqrt{4dr/(1-r)}$ (in which $r$ is the ratio of non-zero coordinates in expectation for the sparsification scheme), the MSE of the ternary stochastic compressor only depends on $\mu$ in the use case of distributed mean estimation (which is the building block of FL). In this sense, we break the three-way tradeoff between communication overhead, privacy, and accuracy by removing the dependency of accuracy on the communication overhead. Different from existing works which suggest that, in the high privacy regime, the error introduced by compression is dominated by the error introduced for privacy, we show that the error caused by compression could be translated into enhancement in privacy. Compared to SQKR [13], the proposed scheme yields better privacy guarantees given the same MSE and communication cost. For the scenario where each user $i$ observes $x_i \in \{-c, c\}^d$ for some constant $c > 0$, the proposed scheme achieves the same privacy guarantee and MSE as those of the classic Gaussian mechanism in the large $d$ regime, which essentially means that the improvement in communication efficiency is achieved for free. We remark that the regime of large $d$ is often of interest in practical FL in which $d$ is the number of training parameters.

## 2   Related Work

Recently, there is a surge of interest in developing differentially private data analysis techniques, which can be divided into three categories: central differential privacy (CDP) that assumes a trusted central server to perturb the collected data [18], distributed differential privacy that relies on secure aggregation during data collection [3], and local differential privacy (LDP) that avoids the need for

the trusted server by perturbing the local data on the user side [19]. To overcome the drawbacks of the Gaussian and Laplacian mechanisms, several discrete mechanisms have been proposed. [18] introduces the one-dimensional binomial noise, which is extended to the general $d$-dimensional case in [6] with more comprehensive analysis in terms of $(\epsilon, \delta)$-DP. [20] analyzes the LDP guarantees of discrete Gaussian noise, while [7] further considers secure aggregation. [8] studies the Rényi DP guarantees of the Skellam mechanism. However, both the discrete Gaussian mechanism and the Skellam mechanism assume infinite ranges at the output, which makes them less communication efficient without appropriate clipping. Moreover, all the above three mechanisms achieve differential privacy at the cost of exploding variance for the additive noise in the high-privacy regimes.

Another line of studies jointly considers privacy preservation and compression. [10, 11] propose to achieve DP by quantizing, sampling, and perturbing each entry, while [12] proposes a vector quantization scheme with local differential privacy. However, the MSE of these schemes grows with $d^2$. [13] investigates the three-way communication-privacy-accuracy tradeoff and incorporates Kashin's representation to achieve order-optimal estimation error in mean estimation. [21] proposes to first sample a portion of coordinates, followed by the randomized response mechanism [22]. [23] and [24] further incorporate shuffling for privacy amplification. [25] proposes to compress the LDP schemes using a pseudorandom generator, while [26] utilizes minimal random coding. [27] proposes a privacy-aware compression mechanism that accommodates DP requirement and unbiasedness simultaneously. However, they consider pure $\epsilon$-DP, which cannot be easily generalized to the relaxed variants. [9] proposes the Poisson binomial mechanism with Rényi DP guarantees. Nonetheless, Rényi DP lacks the favorable hypothesis testing interpretation and the conversion to $(\epsilon, \delta)$-DP is lossy. Moreover, most of the existing works focus on privatizing the compressed data or vice versa, leaving the privacy guarantees of compression mechanisms largely unexplored. [28] proposes a numerical accountant based on fast Fourier transform [29] to evaluate $(\epsilon, \delta)$-DP of general discrete-valued mechanisms. Recently, an independent work [30] studies privacy amplification by compression for central $(\epsilon, \delta)$-DP and multi-message shuffling frameworks. In this work, we consider LDP through the lens of $f$-DP and eliminate the need for a trusted server or shuffler.

Among the relaxations of differential privacy notions [31, 16, 32], $f$-DP [15] is a variant of $\epsilon$-DP with hypothesis testing interpretation, which enjoys the property of lossless conversion to $(\epsilon, \delta)$-DP and tight composition [33]. As a result, it leads to favorable performance in distributed/federated learning [34, 35]. However, to the best of our knowledge, none of the existing works study the $f$-DP of discrete-valued mechanisms. In this work, we bridge the gap by deriving tight $f$-DP guarantees of various compression mechanisms in closed form, based on which a ternary stochastic compressor is proposed to achieve a better communication-privacy-accuracy tradeoff than existing methods.

## 3 Problem Setup and Preliminaries

### 3.1 Problem Setup

We consider a set of $N$ users (denoted by $\mathcal{N}$) with local data $x_i \in \mathbb{R}^d$. The users aim to share $x_i$'s with a central server in a privacy-preserving and communication-efficient manner. More specifically, the users adopt a privacy-preserving mechanism $\mathcal{M}$ to obfuscate their data and share the perturbed results $\mathcal{M}(x_i)$'s with the central server. In the use case of distributed/federated learning, each user has a local dataset $S$. During each training step, it computes the local stochastic gradients and shares the obfuscated gradients with the server. In this sense, the overall gradient computation and obfuscation mechanism $\mathcal{M}$ takes the local dataset $S$ as the input and outputs the obfuscated result $\mathcal{M}(S)$. Upon receiving the shared $\mathcal{M}(S)$'s, the server estimates the mean of the local gradients.

### 3.2 Differential Privacy

Formally, differential privacy is defined as follows.

**Definition 1** (($\epsilon, \delta$)-DP [18]). *A randomized mechanism $\mathcal{M}$ is $(\epsilon, \delta)$-differentially private if for all neighboring datasets $S$ and $S'$ and all $O \subset \mathcal{O}$ in the range of $\mathcal{M}$, we have*
$$P(\mathcal{M}(S) \in O) \leq e^\epsilon P(\mathcal{M}(S') \in O) + \delta, \tag{1}$$
*in which $S$ and $S'$ are neighboring datasets that differ in only one record, and $\epsilon, \delta \geq 0$ are the parameters that characterize the level of differential privacy.*

### 3.3 $f$-Differential Privacy

Assuming that there exist two neighboring datasets $S$ and $S'$, from the hypothesis testing perspective, we have the following two hypotheses

$$H_0 : \text{the underlying dataset is } S, \quad H_1 : \text{the underlying dataset is } S'. \tag{2}$$

Let $P$ and $Q$ denote the probability distribution of $\mathcal{M}(S)$ and $\mathcal{M}(S')$, respectively. [15] formulates the problem of distinguishing the two hypotheses as the tradeoff between the achievable type I and type II error rates. More precisely, consider a rejection rule $0 \leq \phi \leq 1$ (which rejects $H_0$ with a probability of $\phi$), the type I and type II error rates are defined as $\alpha_\phi = \mathbb{E}_P[\phi]$ and $\beta_\phi = 1 - \mathbb{E}_Q[\phi]$, respectively. In this sense, $f$-DP characterizes the tradeoff between type I and type II error rates. The tradeoff function and $f$-DP are formally defined as follows.

**Definition 2** (tradeoff function [15]). *For any two probability distributions $P$ and $Q$ on the same space, the tradeoff function $T(P,Q) : [0,1] \to [0,1]$ is defined as $T(P,Q)(\alpha) = \inf\{\beta_\phi : \alpha_\phi \leq \alpha\}$, where the infimum is taken over all (measurable) rejection rule $\phi$.*

**Definition 3** ($f$-DP [15]). *Let $f$ be a tradeoff function. With a slight abuse of notation, a mechanism $\mathcal{M}$ is $f$-differentially private if $T(\mathcal{M}(S), \mathcal{M}(S')) \geq f$ for all neighboring datasets $S$ and $S'$, which suggests that the attacker cannot achieve a type II error rate smaller than $f(\alpha)$.*

$f$-DP can be converted to $(\epsilon, \delta)$-DP as follows.

**Lemma 1.** *[15] A mechanism is $f(\alpha)$-differentially private if and only if it is $(\epsilon, \delta)$-differentially private with*

$$f(\alpha) = \max\{0, 1 - \delta - e^\epsilon \alpha, e^{-\epsilon}(1 - \delta - \alpha)\}. \tag{3}$$

Finally, we introduce a special case of $f$-DP with $f(\alpha) = \Phi(\Phi^{-1}(1-\alpha) - \mu)$, which is denoted as $\mu$-GDP. More specifically, $\mu$-GDP corresponds to the tradeoff function of two normal distributions with mean 0 and $\mu$, respectively, and a variance of 1.

## 4 Tight $f$-DP Analysis for Existing Discrete-Valued Mechanisms

In this section, we derive the $f$-DP guarantees for a variety of existing differentially private discrete-valued mechanisms in the scalar case (i.e., $d = 1$) to illustrate the main ideas. The vector case will be discussed in Section 6. More specifically, according to Definition 3, the $f$-DP of a mechanism $\mathcal{M}$ is given by the infimum of the tradeoff function over all neighboring datasets $S$ and $S'$, i.e., $f(\alpha) = \inf_{S,S'} \inf_\phi\{\beta_\phi(\alpha) : \alpha_\phi \leq \alpha\}$. Therefore, the analysis consists of two steps: 1) we obtain the closed-form expressions of the tradeoff functions, i.e., $\inf_\phi\{\beta_\phi(\alpha) : \alpha_\phi \leq \alpha\}$, for a generic discrete-valued mechanism (see Section A in the supplementary material); and 2) given the tradeoff functions, we derive the $f$-DP by identifying the mechanism-specific infimums of the tradeoff functions over all possible neighboring datasets. We remark that the tradeoff functions for the discrete-valued mechanisms are essentially piece-wise functions with both the domain and range of each piece determined by both the mechanisms and the datasets, which renders the analysis for the second step highly non-trivial.

### 4.1 Binomial Noise

In this subsection, we consider the binomial noise (i.e., Algorithm 1) proposed in [6], which serves as a communication-efficient alternative to the classic Gaussian noise. More specifically, the output of stochastic quantization in [6] is perturbed by a binomial random variable.

---

**Algorithm 1** Binomial Noise [6]

---

**Input**: $x_i \in [0, 1, \cdots, l]$, $i \in \mathcal{N}$, number of trials $M$, success probability $p$.
Privatization: $Z_i \triangleq x_i + Binom(M, p)$.

---

**Theorem 1.** *Let $\tilde{Z} = Binom(M, p)$, the binomial noise mechanism in Algorithm 1 is $f^{bn}(\alpha)$-differentially private with*

$$f^{bn}(\alpha) = \min\{\beta^+_{\phi, \inf}(\alpha), \beta^-_{\phi, \inf}(\alpha)\}, \tag{4}$$

*in which*

$$\beta_{\phi,\inf}^{+}(\alpha) = \begin{cases} P(\tilde{Z} \geq \tilde{k}+l) + \frac{P(Z=\tilde{k}+l)P(\tilde{Z}<\tilde{k})}{P(\tilde{Z}=\tilde{k})} - \frac{P(\tilde{Z}=\tilde{k}+l)}{P(\tilde{Z}=\tilde{k})}\alpha, \\ \qquad\qquad for\ \alpha \in [P(\tilde{Z} < \tilde{k}), P(\tilde{Z} \leq \tilde{k})], \tilde{k} \in [0, M-l], \\ 0, \qquad\qquad\qquad\qquad\qquad\qquad for\ \alpha \in [P(\tilde{Z} \leq M-l), 1]. \end{cases}$$

$$(5)$$

$$\beta_{\phi,\inf}^{-}(\alpha) = \begin{cases} P(\tilde{Z} \leq \tilde{k}-l) + \frac{P(\tilde{Z}=\tilde{k}-l)P(\tilde{Z}>\tilde{k})}{P(\tilde{Z}=\tilde{k})} - \frac{P(\tilde{Z}=\tilde{k}-l)}{P(\tilde{Z}=\tilde{k})}\alpha, \\ \qquad\qquad for\ \alpha \in [P(\tilde{Z} > \tilde{k}), P(\tilde{Z} \geq \tilde{k})], \tilde{k} \in [l, M], \qquad (6) \\ 0, \qquad\qquad\qquad\qquad\qquad\qquad for\ \alpha \in [P(\tilde{Z} \geq l), 1]. \end{cases}$$

*Given that* $P(\tilde{Z} = k) = \binom{M}{k}p^k(1-p)^{M-k}$, *it can be readily shown that when* $p = 0.5$*, both* $\beta_{\phi,\inf}^{+}(\alpha)$ *and* $\beta_{\phi,\inf}^{-}(\alpha)$ *are maximized, and* $f(\alpha) = \beta_{\phi,\inf}^{+}(\alpha) = \beta_{\phi,\inf}^{-}(\alpha)$*.*

Fig. 1 shows the impact of $M$ when $l = 8$, which confirms the result in [6] that a larger $M$ provides better privacy protection (recall that given the same $\alpha$, a larger $\beta_\alpha$ indicates that the attacker makes mistakes in the hypothesis testing more likely and therefore corresponds to better privacy protection). Note that the output of Algorithm 1 $Z_i \in \{0, 1, ..., M + l\}$, which requires a communication overhead of $\log_2(M+l+1)$ bits. We can readily convert $f(\alpha)$-DP to $(\epsilon, \delta)$-DP by utilizing Lemma 1.

**Remark 1.** *The results derived in this work improve [6] in two aspects: (1) Theorem 1 in [6] requires* $Mp(1 - p) \geq \max(23\log(10d/\delta), 2l/s) > \max(23\log(10), 2l/s)$*, in which* $1/s \in \mathbb{N}$ *is some scaling factor. When* $p = 1/2$*, it requires* $M \geq 212$*. More specifically, for* $M = 500$*, [6] requires* $\delta > 0.044$*. Our results imply that there exists some* $(\epsilon, \delta)$ *such that Algorithm 1 is* $(\epsilon, \delta)$*-DP as long as* $M > l$*. For* $M = 500$*,* $\delta$ *can be as small as* $4.61 \times 10^{-136}$*. (2) Our results are tight, in the sense that no relaxation is applied in our derivation. As an example, when* $M = 500$ *and* $p = 0.5$*, Theorem 1 in [6] gives* $(3.18, 0.044)$*-DP while Theorem 1 in this paper yields* $(1.67, 0.039)$*-DP.*

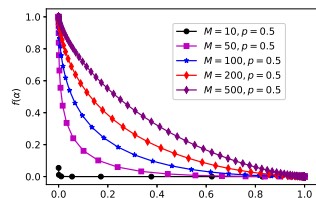

Figure 1: Impact of $M$ on Algorithm 1 with $l = 8$.

## 4.2 Binomial Mechanism

---
**Algorithm 2** Binomial Mechanism [9]

---
**Input**: $c > 0$, $x_i \in [-c, c]$, $M \in \mathbb{N}$, $p_i(x_i) \in [p_{min}, p_{max}]$
Privatization: $Z_i \triangleq Binom(M, p_i(x_i))$.

---

In this subsection, we consider the binomial mechanism (i.e., Algorithm 2). Different from Algorithm 1 that perturbs the data with noise following the binomial distribution with the same success probability, the binomial mechanism encodes the input $x_i$ into the success probability of the binomial distribution. We establish the privacy guarantee of Algorithm 2 as follows.

**Theorem 2.** *The binomial mechanism in Algorithm 2 is* $f^{bm}(\alpha)$*-differentially private with*
$$f^{bm}(\alpha) = \min\{\beta_{\phi,\inf}^{+}(\alpha), \beta_{\phi,\inf}^{-}(\alpha)\}, \qquad (7)$$
*in which*

$$\beta_{\phi,\inf}^{+}(\alpha) = 1 - [P(Y < k) + \gamma P(Y = k)] = P(Y \geq k) + \frac{P(Y = k)P(X < k)}{P(X = k)} - \frac{P(Y = k)}{P(X = k)}\alpha,$$

*for* $\alpha \in [P(X < k), P(X \leq k)]$ *and* $k \in \{0, 1, 2, \cdots, M\}$*, where* $X = Binom(M, p_{max})$ *and* $Y = Binom(M, p_{min})$*, and*

$$\beta_{\phi,\inf}^{-}(\alpha) = 1 - [P(Y > k) + \gamma P(Y = k)] = P(Y \leq k) + \frac{P(Y = k)P(X > k)}{P(X = k)} - \frac{P(Y = k)}{P(X = k)}\alpha,$$

*for* $\alpha \in [P(X > k), P(X \geq k)]$ *and* $k \in \{0, 1, 2, \cdots, M\}$*, where* $X = Binom(M, p_{min})$ *and* $Y = Binom(M, p_{max})$*. When* $p_{max} = 1 - p_{min}$*, we have* $\beta_{\phi,\inf}^{+}(\alpha) = \beta_{\phi,\inf}^{-}(\alpha)$*.*

**Remark 2** (**Comparison to [9]**)**.** *The binomial mechanism is part of the Poisson binomial mechanism proposed in [9]. More specifically, in [9], each user $i$ shares the output of the binomial mechanism*

$Z_i$ with the server, in which $p_i(x_i) = \frac{1}{2} + \frac{\theta}{c}x_i$ and $\theta$ is some design parameter. It can be readily verified that $p_{max} = 1 - p_{min}$ in this case. The server then aggregates the result through $\bar{x} = \frac{c}{MN\theta}(\sum_{i \in \mathcal{N}} Z_i - \frac{MN}{2})$. [9] requires secure aggregation and considers the privacy leakage of releasing $\bar{x}$, while we complement it by showing the LDP, i.e., the privacy leakage of releasing $Z_i$ for each user. In addition, we eliminate the constraint $\theta \in [0, \frac{1}{4}]$, and the results hold for any selection of $p_i(x_i)$. Moreover, the privacy guarantees in Theorem 2 are tight since no relaxation is involved. Fig. 2 shows the impact of $M$ on the privacy guarantee. In contrast to binomial noise, the privacy of the binomial mechanisms improves as $M$ (and equivalently communication overhead) decreases, which implies that it is more suitable for communication-constrained scenarios. We also derive the $f$-DP of the Poisson binomial mechanism, which are presented in Section C in the supplementary material.

In the following, we present two existing compressors that are special cases of the binomial mechanism.

**Example 1.** *We first consider the following stochastic sign compressor proposed in [36].*

**Definition 4** (**Two-Level Stochastic Compressor** [36]). *For any given $x \in [-c, c]$, the compressor sto-sign outputs*

$$sto\text{-}sign(x, A) = \begin{cases} 1, & \text{with probability } \frac{A+x}{2A}, \\ -1, & \text{with probability } \frac{A-x}{2A}, \end{cases} \quad (8)$$

*where $A > c$ is the design parameter that controls the level of stochasticity.*

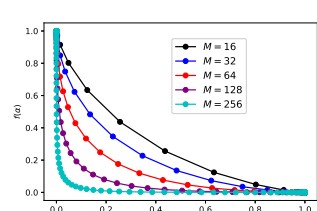

Figure 2: Impact of $M$ on Algorithm 2.

*With a slight modification (i.e., mapping the output space from $\{0, 1\}$ to $\{-1, 1\}$), sto-sign$(x, A)$ can be understood as a special case of the binomial mechanism with $M = 1$ and $p_i(x_i) = \frac{A+x_i}{2A}$. In this case, we have $p_{max} = \frac{A+c}{2A}$ and $p_{min} = \frac{A-c}{2A}$. Applying the results in Theorem 2 yields*

$$f^{sto\text{-}sign}(\alpha) = \beta^+_{\phi,\inf}(\alpha) = \beta^-_{\phi,\inf}(\alpha) = \begin{cases} 1 - \frac{A+c}{A-c}\alpha, & \text{for } \alpha \in [0, \frac{A+c}{2A}], \\ \frac{A-c}{A+c} - \frac{A-c}{A+c}\alpha, & \text{for } \alpha \in [\frac{A+c}{2A}, 1]. \end{cases} \quad (9)$$

*Combining (9) with (3) suggests that the sto-sign compressor ensures $(\ln(\frac{A+c}{A-c}), 0)$-DP.*

**Example 2.** *The second sign-based compressor that we examine is $CLDP_\infty(\cdot)$ [23].*

**Definition 5** ($CLDP_\infty(\cdot)$ [23]). *For any given $x \in [-c, c]$, the compressor $CLDP_\infty(\cdot)$ outputs $CLDP_\infty(\epsilon)$, which is given by*

$$CLDP_\infty(\epsilon) = \begin{cases} +1, & \text{with probability } \frac{1}{2} + \frac{x}{2c}\frac{e^\epsilon - 1}{e^\epsilon + 1}, \\ -1, & \text{with probability } \frac{1}{2} - \frac{x}{2c}\frac{e^\epsilon - 1}{e^\epsilon + 1}. \end{cases} \quad (10)$$

$CLDP_\infty(\epsilon)$ *can be understood as a special case of sto-sign$(x, A)$ with $A = \frac{c(e^\epsilon + 1)}{e^\epsilon - 1}$. In this case, according to (9), we have*

$$f^{CLDP_\infty}(\alpha) = \begin{cases} 1 - e^\epsilon\alpha, & \text{for } \alpha \in [0, \frac{A+c}{2A}], \\ e^{-\epsilon}(1 - \alpha), & \text{for } \alpha \in [\frac{A+c}{2A}, 1]. \end{cases} \quad (11)$$

*Combining the above result with (3) suggests that $CLDP_\infty(\epsilon)$ ensures $(\epsilon, 0)$-DP, which recovers the result in [23]. It is worth mentioning that $CLDP_\infty(\epsilon)$ can be understood as the composition of sto-sign with $A = c$ followed by the randomized response mechanism [22], and is equivalent to the one-dimensional case of the compressor in [13]. Moreover, the one-dimensional case of the schemes in [10, 11] can also be understood as special cases of sto-sign.*

## 5  The Proposed Ternary Compressor

The output of the binomial mechanism with $M = 1$ lies in the set $\{0, 1\}$, which coincides with the sign-based compressor. In this section, we extend the analysis to the ternary case, which can be understood as a combination of sign-based quantization and sparsification (when the output takes value 0, no transmission is needed since it does not contain any information) and leads to improved communication efficiency. More specifically, we propose the following ternary compressor.

**Definition 6** (**Ternary Stochastic Compressor**). *For any given $x \in [-c, c]$, the compressor $ternary$ outputs $ternary(x, A, B)$, which is given by*

$$ternary(x, A, B) = \begin{cases} 1, & \text{with probability } \frac{A+x}{2B}, \\ 0, & \text{with probability } 1 - \frac{A}{B}, \\ -1, & \text{with probability } \frac{A-x}{2B}, \end{cases} \quad (12)$$

*where $B > A > c$ are the design parameters that control the level of sparsity.*

For the ternary stochastic compressor in Definition 6, we establish its privacy guarantee as follows.

**Theorem 3.** *The ternary stochastic compressor is $f^{ternary}(\alpha)$-differentially private with*

$$f^{ternary}(\alpha) = \begin{cases} 1 - \frac{A+c}{A-c}\alpha, & \text{for } \alpha \in [0, \frac{A-c}{2B}], \\ 1 - \frac{c}{B} - \alpha, & \text{for } \alpha \in [\frac{A-c}{2B}, 1 - \frac{A+c}{2B}], \\ \frac{A-c}{A+c} - \frac{A-c}{A+c}\alpha, & \text{for } \alpha \in [1 - \frac{A+c}{2B}, 1]. \end{cases} \quad (13)$$

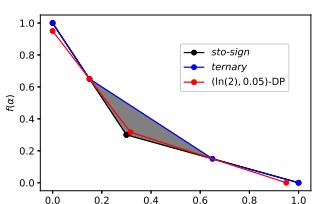

Figure 3: Sparsification improves privacy.

**Remark 3** (Privacy amplification by sparsification). *It can be observed from (9) and (13) that $f^{ternary}(\alpha) > f^{sto\text{-}sign}$ when $\alpha \in [\frac{A-c}{2B}, 1 - \frac{A+c}{2B}]$, and $f^{ternary}(\alpha) = f^{sto\text{-}sign}$, otherwise. Fig. 3 shows $f^{ternary}(\alpha)$ and $f^{sto\text{-}sign}$ for $c = 0.1, A = 0.25, B = 0.5$, and the shaded gray area corresponds to the improvement in privacy. It can be observed that communication efficiency and privacy are improved simultaneously. It is worth mentioning that, if we convert the privacy guarantees to $(\epsilon, 0)$-DP, we have $\epsilon = \ln(\frac{7}{3})$ for both compressors. However, the ternary compressor ensures $(\ln(2), 0.05)$-DP (i.e., $f^{ternary}(\alpha) \geq \max\{0, 0.95 - 2\alpha, 0.5(0.95 - \alpha)\}$) while the $sto\text{-}sign$ compressor does not. We note that for the same $A$, as $B$ increases (i.e., communication cost decreases), $f^{ternary}(\alpha)$ approaches $f(\alpha) = 1 - \alpha$ (which corresponds to perfect privacy).*

In the following, we present a special case of the proposed ternary stochastic compressor.

**Example 3.** *The ternary-based compressor proposed in [37] is formally defined as follows.*

**Definition 7** ($ternarize(\cdot)$ [37]). *For any given $x \in [-c, c]$, the compressor $ternarize(\cdot)$ outputs $ternarize(x, B) = sign(x)$ with probability $|x|/B$ and $ternarize(x, B) = 0$ otherwise, in which $B > c$ is the design parameter.*

*$ternarize(x, B)$ can be understood as a special case of $ternary(x, A, B)$ with $A = |x|$. According to Theorem 3, $f^{ternary}(\alpha) = 1 - \frac{c}{B} - \alpha$ for $\alpha \in [0, 1 - \frac{c}{B}]$ and $f^{ternary}(\alpha) = 0$ for $\alpha \in [1 - \frac{c}{B}, 1]$. Combining the above result with (3), we have $\delta = \frac{c}{B}$ and $\epsilon = 0$, i.e., $ternarize(\cdot)$ provides perfect privacy protection ($\epsilon = 0$) with a violation probability of $\delta = \frac{c}{B}$. Specifically, the attacker cannot distinguish $x_i$ from $x_i'$ if the output of $ternarize(\cdot) = 0$ (perfect privacy protection), while no differential privacy is provided if the output of $ternarize(\cdot) \neq 0$ (violation of the privacy guarantee).*

**Remark 4.** *It is worth mentioning that, in [37], the users transmit a scaled version of $ternarize(\cdot)$ and the scaling factor reveals the magnitude information of $x_i$. Therefore, the compressor in [37] is not differentially private.*

## 6 Breaking the Communication-Privacy-Accuracy Tradeoff

In this section, we extend the results in Section 5 to the vector case in two different approaches, followed by discussions on the three-way tradeoff between communication, privacy, and accuracy. The results in Section 4 can be extended similarly. Specifically, in the first approach, we derive the $\mu$-GDP in closed form, while introducing some loss in privacy guarantees. In the second approach, a tight approximation is presented. Given the results in Section 5, we can readily convert $f$-DP in the scalar case to Gaussian differential privacy in the vector case as follows.

**Theorem 4.** *Given a vector $x_i = [x_{i,1}, x_{i,2}, \cdots, x_{i,d}]$ with $|x_{i,j}| \leq c, \forall j$. Applying the ternary compressor to the $j$-th coordinate of $x_i$ independently yields $\mu$-GDP with $\mu = -2\Phi^{-1}(\frac{1}{1+(\frac{A+c}{A-c})^d})$.*

**Remark 5.** *Note that $\|x_i\|_2 \leq c$ is a sufficient condition for $|x_{i,j}| \leq c, \forall j$. In the proof of Theorem 4, we first convert $f^{ternary}(\alpha)$-DP to $(\epsilon, 0)$-DP for the scalar case, and then obtain $(d\epsilon, 0)$-DP*

*for the d-dimensional case, followed by the conversion to GDP. One may notice that some loss in privacy guarantee is introduced since the extreme case $|x_{i,j}| = c, \forall j$ actually violates the condition $||x_i||_2 \leq c$. To address this issue, following a similar method in [13, 38, 9], one may introduce Kashin's representation to transform the $l_2$ geometry of the data into the $l_\infty$ geometry. More specifically, [39] shows that for $D > d$, there exists a tight frame $U$ such that for any $x \in \mathbb{R}^d$, one can always represent each $x_i$ with $y_i \in [-\gamma_0/\sqrt{d}, -\gamma_0/\sqrt{d}]^D$ for some $\gamma_0$ and $x_i = Uy_i$.*

In Theorem 4, some loss in privacy guarantees is introduced when we convert $f$-DP to $\mu$-GDP. In fact, since each coordinate of the vector is processed independently, the extension from the scalar case to the $d$-dimensional case may be understood as the $d$-fold composition of the mechanism in the scalar case. The composed result can be well approximated or numerically obtained via the central limit theorem for $f$-DP in [15] or the Edgeworth expansion in [33]. In the following, we present the result for the ternary compressor by utilizing the central limit theorem for $f$-DP.

**Theorem 5.** *For a vector $x_i = [x_{i,1}, x_{i,2}, \cdots, x_{i,d}]$ with $|x_{i,j}| \leq c, \forall j$, the ternary compressor with $B \geq A > c$ is $f^{ternary}(\alpha)$-DP with*

$$G_\mu(\alpha + \gamma) - \gamma \leq f^{ternary}(\alpha) \leq G_\mu(\alpha - \gamma) + \gamma, \tag{14}$$

*in which*

$$\mu = \frac{2\sqrt{d}c}{\sqrt{AB - c^2}}, \quad \gamma = \frac{0.56 \left[ \frac{A-c}{2B} \left| 1 + \frac{c}{B} \right|^3 + \frac{A+c}{2B} \left| 1 - \frac{c}{B} \right|^3 + \left( 1 - \frac{A}{B} \right) \left| \frac{c}{B} \right|^3 \right]}{(\frac{A}{B} - \frac{c^2}{B^2})^{3/2} d^{1/2}}. \tag{15}$$

Given the above results, we investigate the communication-privacy-accuracy tradeoff and compare the proposed ternary stochastic compressor with the state-of-the-art method SQKR in [13] and the classic Gaussian mechanism. According to the discussion in Remark 5, given the $l_2$ norm constraint, Kashin's representation can be applied to transform it into the $l_\infty$ geometry. Therefore, for ease of discussion, we consider the setting in which each user $i$ stores a vector $x_i = [x_{i,1}, x_{i,2}, \cdots, x_{i,d}]$ with $|x_{i,j}| \leq c = \frac{C}{\sqrt{d}}, \forall j$, and $||x_i||_2 \leq C$.

**Ternary Stochastic Compressor**: Let $Z_{i,j} = ternary(x_{i,j}, A, B)$, then $\mathbb{E}[BZ_{i,j}] = x_{i,j}$ and $Var(BZ_{i,j}) = AB - x_{i,j}^2$. In this sense, applying the ternary stochastic compressor to each coordinate of $x_i$ independently yields an unbiased estimator with a variance of $ABd - ||x_i||_2^2$. The privacy guarantee is given by Theorem 5, and the communication overhead is $(\log_2(d) + 1)\frac{A}{B}d$ bits in expectation.

**SQKR**: In SQKR, each user first quantizes each coordinate of $x_i$ to $\{-c, c\}$ with 1-bit stochastic quantization. Then, it samples $k$ coordinates (with replacement) and privatizes the $k$ bit message via the $2^k$ Random response mechanism with $\epsilon$-LDP [14]. The SQKR mechanism yields an unbiased estimator with a variance of $\frac{d}{k}(\frac{e^\epsilon + 2^k - 1}{e^\epsilon - 1})^2 C^2 - ||x_i||_2^2$. The privacy guarantee is $\epsilon$-LDP, and the corresponding communication overhead is $(\log_2(d) + 1)k$ bits.

**Gaussian Mechanism**: We apply the Gaussian mechanism (i.e., adding independent zero-mean Gaussian noise $n_{i,j} \sim \mathcal{N}(0, \sigma^2)$ to $x_{i,j}$), followed by a sparsification probability of $1 - A/B$ as in $ternary(x_{i,j}, A, B)$, which gives $Z_{i,j}^{Gauss} = \frac{B}{A}(x_{i,j} + n_{i,j})$ with probability $A/B$ and $Z_{i,j}^{Gauss} = 0$, otherwise. It can be observed that $\mathbb{E}[Z_{i,j}^{Gauss}] = x_{i,j}$ and $Var(Z_{i,j}^{Gauss}) = \frac{B}{A}\sigma^2 + (\frac{B}{A} - 1)x_{i,j}^2$. Therefore, the Gaussian mechanism yields an unbiased estimator with a variance of $\frac{B}{A}\sigma^2 d + (\frac{B}{A} - 1)||x_i||_2^2$. By utilizing the post-processing property, it can be shown that the above Gaussian mechanism is $\frac{2\sqrt{d}c}{\sigma}$-GDP [15], and the communication overhead is $(\log_2(d) + 32)\frac{A}{B}d$ bits in expectation.

**Discussion**: It can be observed that for SQKR, with a given privacy guarantee $\epsilon$-LDP, the variance (i.e., MSE) depends on $k$ (i.e., the communication overhead). When $e^\epsilon \ll 2^k$ (which corresponds to the high privacy regime), the variance grows rapidly as $k$ increases. For the proposed ternary stochastic compressor, it can be observed that both the privacy guarantee (in terms of $\mu$-GDP) and the variance depend on $AB$. Particularly, with a given privacy guarantee $\mu < \sqrt{4dr/(1 - r)}$ for $r = A/B$, the variance is given by $(4d/\mu^2 + 1)C^2 - ||x_i||_2^2$, which remains the same regardless of the communication overhead. **In this sense, we essentially remove the dependency of accuracy on the communication overhead and therefore break the three-way tradeoff between communication**

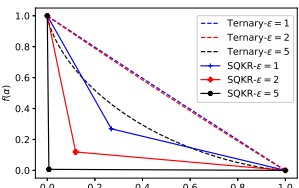 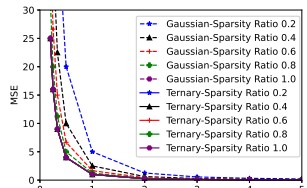 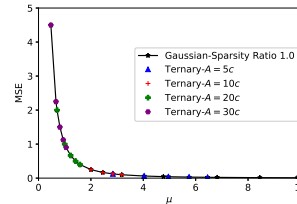

Figure 4: For the left figure, we set $k = 10$ and derive the corresponding variance for SQKR, based on which $A$ and $B$ for the ternary stochastic compressor are computed such that they have the same communication overhead and MSE in expectation. The middle and right figures show the tradeoff between $\mu$-GDP and MSE. For the middle figure, we set $\sigma \in \{\frac{2}{5}, \frac{1}{2}, \frac{2}{3}, 1, 2, 4, 6, 8, 10\}$ for the Gaussian mechanism, given which $A$ and $B$ are computed such that $AB = c^2 + \sigma^2$ and the sparsity ratio is $A/B$. For the right figure, we set $A \in \{5c, 10c, 20c, 30c\}$ and $A/B \in \{0.2, 0.4, 0.6, 0.8, 1.0\}$, given which the corresponding $\sigma$'s are computed such that $AB = c^2 + \sigma^2$.

**overhead, privacy, and accuracy.**[1] This is mainly realized by accounting for privacy amplification by sparsification. At a high level, when fewer coordinates are shared (which corresponds to a larger privacy amplification and a larger MSE), the ternary stochastic compressor introduces less ambiguity to each coordinate (which corresponds to worse privacy protection and a smaller MSE) such that both the privacy guarantee and the MSE remain the same. Since we use different differential privacy measures from [13] (i.e., $\mu$-GDP in this work and $\epsilon$-DP in [13]), we focus on the comparison between the proposed ternary stochastic compressor and the Gaussian mechanism (which is order-optimal in most parameter regimes, see [30]) in the following discussion and present the detailed comparison with SQKR in the experiments in Section 7.

Let $AB = c^2 + \sigma^2$, it can be observed that the $f$-DP guarantee of the ternary compressor approaches that of the Gaussian mechanism as $d$ increases, and the corresponding variance is given by $Var(BZ_{i,j}) = \sigma^2 + c^2 - x_{i,j}^2$. When $A = B$, i.e., no sparsification is applied, we have $Var(BZ_{i,j}) - Var(Z_{i,j}^{Gauss}) = c^2 - x_{i,j}^2$. Specifically, when $x_{i,j} \in \{-c, c\}, \forall 1 \leq j \leq d$, the ternary compressor demonstrates the same $f$-DP privacy guarantee and variance as that for the Gaussian mechanism, i.e., **the improvement in communication efficiency is obtained for free (in the large $d$ regime)**. When $B > A$, we have $Var(BZ_{i,j}) - Var(Z_{i,j}^{Gauss}) = (1 - \frac{B}{A})\sigma^2 + c^2 - \frac{B}{A}x_{i,j}^2$, and there exists some $B$ such that the ternary compressor outperforms the Gaussian mechanism in terms of both variance and communication efficiency. It is worth mentioning that the privacy guarantee of the Gaussian mechanism is derived by utilizing the post-processing property. We believe that sparsification brings improvement in privacy for the Gaussian mechanism as well, which is, however, beyond the scope of this paper.

**Optimality:** It has been shown that, for $k$-bit unbiased compression mechanisms, there is a lower bound of $\Omega(C^2 d/k)$ in MSE [40]. For the proposed ternary compressor, the MSE and the communication cost are given by $O(ABd)$ and $A(\log(d)+1)d/B$ bits, respectively. Let $k = A(\log(d)+1)d/B$, it achieves an MSE of $O(A^2 d^2 (\log(d) + 1)/k)$. Since $A > c = C/\sqrt{d}$, the MSE of the ternary compressor is given by $O(C^2 d(\log(d)+1)/k)$, which implies that it is order-optimal up to a factor of $\log(d)$. Note that the factor of $\log(d)$ is used to represent the indices of coordinates that are non-zero, which can be eliminated by allowing for shared randomness between the users and the server.

## 7 Experiments

In this section, we examine the performance of the proposed ternary compressor in the case of distributed mean estimation. We follow the set-up of [9] and generate $N = 1000$ user vectors with dimension $d = 250$, i.e., $x_1, ..., x_N \in \mathbb{R}^{250}$. Each local vector has bounded $l_2$ and $l_\infty$ norms, i.e., $||x_i||_2 \leq C = 1$ and $||x_i||_\infty \leq c = \frac{1}{\sqrt{d}}$.

Fig. 4 compares the proposed ternary stochastic compressor with SQKR and the Gaussian mechanism. More specifically, the left figure in Fig. 4 compares the privacy guarantees (in terms of the tradeoff between type I and type II error rates) of the ternary stochastic compressor and SQKR given the

---

[1]In practice, utilizing the closed-form expressions of the MSE and the privacy guarantee $\mu$, one may readily obtain the corresponding $A$ and $B$ for any given privacy/MSE and communication cost specifications.

same communication overhead and MSE. It can be observed that the proposed ternary stochastic compressor outperforms SQKR in terms of privacy preservation, i.e., given the same type I error rate $\alpha$, the type II error rate $\beta$ of the ternary stochastic compressor is significantly larger than that of SQKR, which implies better privacy protection. For example, for SQKR with $\epsilon = 2$, given type I error rate $\alpha = 0.5$, the type II error rate of the attacker is around $f^{SQKR}(\alpha) = 0.068$, while the ternary compressor attains $f^{ternary}(\alpha) = 0.484$. Given the same MSE and communication cost as that of SQKR with $\epsilon_{SQKR} = \{1, 2, 5\}$, if we translate the privacy guarantees of the ternary compressor from $f$-DP to $\epsilon$-DP via Lemma 1 (we numerically test different $\epsilon$'s such that $f^{ternary}(\alpha) \geq \max\{0, 1 - \delta - e^\epsilon\alpha, e^{-\epsilon}(1 - \delta - \alpha)\}$ holds for $\delta = 0$), we have $\epsilon_{ternary} = \{0.05, 0.2, 3.9\}$ for the ternary compressor, which demonstrates its effectiveness. The middle and right figures in Fig. 4 show the tradeoff between MSE and DP guarantees for the Gaussian mechanism and the proposed ternary compressor. Particularly, in the middle figure, the tradeoff curves for the ternary compressor with all the examined sparsity ratios overlap with that of the Gaussian mechanism with $A/B = 1$ since they essentially have the same privacy guarantees, and the difference in MSE is negligible. For the Gaussian mechanism with $\frac{A}{B} < 1$, the MSE is larger due to sparsification, which validates our discussion in Section 6. In the right figure, we examine the MSEs of the proposed ternary compressor with various $A$'s and $B$'s. It can be observed that the corresponding tradeoff between MSE and privacy guarantee matches that of the Gaussian mechanism well, which validates that the improvement in communication efficiency for the proposed ternary compressor is obtained for free.

## 8    Limitation

The main results derived in this paper are for the scalar case, which are extended to the vector case by invoking the central limit theorem. In this case, the privacy guarantees derived in Theorem 5 are tight only in the large $d$ regime. Fortunately, in applications like distributed learning, $d$ corresponds to the model size (usually in the orders of millions for modern neural networks). Moreover, despite that the privacy-accuracy tradeoff of the proposed ternary compressor matches that of the Gaussian mechanism which is order-optimal in $(\epsilon, \delta)$-DP, the optimality of the proposed ternary compressor in the $f$-DP regime needs to be further established.

## 9    Conclusion

In this paper, we derived the privacy guarantees of discrete-valued mechanisms with finite output space in the lens of $f$-differential privacy, which covered various differentially private mechanisms and compression mechanisms as special cases. Through leveraging the privacy amplification by sparsification, a ternary compressor that achieves better accuracy-privacy-communication tradeoff than existing methods is proposed. It is expected that the proposed methods can find broader applications in the design of communication efficient and differentially private federated data analysis techniques.

## Acknowledgments and Disclosure of Funding

Richeng Jin was supported in part by the National Natural Science Foundation of China under Grant No. 62301487, in part by the Zhejiang Provincial Natural Science Foundation of China under Grant No. LQ23F010021, and in part by the Ng Teng Fong Charitable Foundation in the form of ZJU-SUTD IDEA Grant No. 188170-11102. Zhonggen Su was supported by the Fundamental Research Funds for the Central Universities Grants. Zhaoyang Zhang was supported in part by the National Natural Science Foundation of China under Grant No. U20A20158, in part by the National Key R&D Program of China under Grant No. 2020YFB1807101, and in part by the Zhejiang Provincial Key R&D Program under Grant No. 2023C01021. Huaiyu Dai was supported by the US National Science Foundation under Grant No. ECCS-2203214. The views expressed in this publication are those of the authors and do not necessarily reflect the views of the National Science Foundation.

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

# Breaking the Communication-Privacy-Accuracy Tradeoff with $f$-Differential Privacy: Supplementary Material

## A   Tradeoff Functions for a Generic Discrete-Valued Mechanism

We consider a general randomization protocol $\mathcal{M}(\cdot)$ with discrete and finite output space. In this case, we can always find a one-to-one mapping between the range of $\mathcal{M}(\cdot)$ and a subset of $\mathbb{Z}$. With such consideration, we assume that the output of the randomization protocol is an integer, i.e., $\mathcal{M}(S) \in \mathbb{Z}_\mathcal{M} \subset \mathbb{Z}, \forall S$, without loss of generality. Given the randomization protocol and the hypothesis testing problem in (2), we derive its tradeoff function as a function of the type I error rate in the following lemma.

**Lemma 2.** *For two neighboring datasets $S$ and $S'$, suppose that the range of the randomized mechanism $\mathcal{R}(\mathcal{M}(S)) \cup \mathcal{R}(\mathcal{M}(S')) = \mathbb{Z}_\mathcal{M}^U = [\mathcal{Z}_L^U, \ldots, \mathcal{Z}_R^U] \subset \mathbb{Z}$ and $\mathcal{R}(\mathcal{M}(S)) \cap \mathcal{R}(\mathcal{M}(S')) = \mathbb{Z}_\mathcal{M}^I = [\mathcal{Z}_L^I, \ldots, \mathcal{Z}_R^I] \subset \mathbb{Z}$. Let $X = \mathcal{M}(S)$ and $Y = \mathcal{M}(S')$. Then,*

*Case (1) If $\mathcal{M}(S) \in [\mathcal{Z}_L^I, \mathcal{Z}_L^I + 1, \ldots, \mathcal{Z}_R^U]$, $\mathcal{M}(S') \in [\mathcal{Z}_L^U, \mathcal{Z}_L^U + 1, \ldots, \mathcal{Z}_R^I]$, and $\frac{P(Y=k)}{P(X=k)}$ is a decreasing function of $k$ for $k \in \mathbb{Z}_\mathcal{M}^I$, the tradeoff function in Definition 2 is given by*

$$\beta_\phi^+(\alpha) = \begin{cases} P(Y \geq k) + \frac{P(Y=k)P(X<k)}{P(X=k)} - \frac{P(Y=k)}{P(X=k)}\alpha, \\ \qquad \text{if } \alpha \in (P(X < k), P(X \leq k)], \ k \in [\mathcal{Z}_L^I, \mathcal{Z}_R^I]. \\ 0, \qquad\qquad\qquad\qquad \text{if } \alpha \in (P(X < \mathcal{Z}_R^I + 1), 1]. \end{cases} \tag{16}$$

*Case (2) If $\mathcal{M}(S) \in [\mathcal{Z}_L^U, \mathcal{Z}_L^U + 1, \cdots, \mathcal{Z}_R^I]$, $\mathcal{M}(S') \in [\mathcal{Z}_L^I, \mathcal{Z}_L^I + 1, \cdots, \mathcal{Z}_R^U]$, and $\frac{P(Y=k)}{P(X=k)}$ is an increasing function of $k$ for $k \in \mathbb{Z}_\mathcal{M}^I$, the tradeoff function in Definition 2 is given by*

$$\beta_\phi^-(\alpha) = \begin{cases} P(Y \leq k) + \frac{P(Y=k)P(X>k)}{P(X=k)} - \frac{P(Y=k)}{P(X=k)}\alpha, \\ \qquad \text{if } \alpha \in (P(X > k), P(X \geq k)], \ k \in [\mathcal{Z}_L^I, \mathcal{Z}_R^I]. \\ 0, \qquad\qquad\qquad\qquad \text{if } \alpha \in (P(X > \mathcal{Z}_L^I - 1), 1]. \end{cases} \tag{17}$$

**Remark 6.** *It is assumed in Lemma 2 that $\frac{P(Y=k)}{P(X=k)}$ is a decreasing function (for part (1)) or an increasing function (for part (2)) of $k \in \mathbb{Z}_\mathcal{M}^I$, without loss of generality. In practice, thanks to the post-processing property of DP [15], one can relabel the output of the mechanism to ensure that this condition holds and Lemma 2 can be adapted accordingly.*

**Remark 7.** *We note that in Lemma 2, both $X$ and $Y$ depend on both the randomized mechanism $\mathcal{M}(\cdot)$ and the neighboring datasets $S$ and $S'$. Therefore, the infimums of the tradeoff functions in (16) and (17) are mechanism-specific, which should be analyzed individually. After identifying the neighboring datasets $S$ and $S'$ that minimize $\beta_\phi^+(\alpha)$ and $\beta_\phi^-(\alpha)$ for a mechanism $\mathcal{M}(\cdot)$ (which is highly non-trivial), we can obtain the distributions of $X$ and $Y$ in (16) and (17) and derive the corresponding $f$-DP guarantees.*

**Remark 8.** *Since $\beta_\phi^+(\alpha)$ is a piecewise function with decreasing slopes w.r.t $k$ (see, e.g., Fig. 1), it can be readily shown that $\beta_\phi^+(\alpha) \geq \max\{P(Y \geq k) + \frac{P(Y=k)}{P(X=k)}P(X < k) - \frac{P(Y=k)}{P(X=k)}\alpha, 0\}, \forall k \in \mathbb{Z}_\mathcal{M}^I$. As a result, utilizing Lemma 1, we may obtain different pairs of $(\epsilon, \delta)$ given different $k$'s.*

**Remark 9.** *Although we assume a finite output space, a similar method can be applied to the mechanisms with an infinite range. Taking the discrete Gaussian noise [20] as an example, $\mathcal{M}(x) = x + V$ with $P(V = v) = \frac{e^{-v^2/2\sigma^2}}{\sum_{v \in \mathbb{Z}} e^{-v^2/2\sigma^2}}$. One may easily verify that $\frac{P(\mathcal{M}(x_i)=k)}{P(\mathcal{M}(x_i')=k)}$ is a decreasing function of $k$ if $x_i' > x_i$ (and increasing otherwise). Then we can find some threshold $v$ for the rejection rule $\phi$ such that $\alpha_\phi = P(\mathcal{M}(x_i) \leq v) = \alpha$, and the corresponding $\beta_\phi(\alpha) = 1 - P(\mathcal{M}(x_i') \leq v)$.*

The key to proving Lemma 2 is finding the rejection rule $\phi$ such that $\beta_\phi(\alpha)$ is minimized for a pre-determined $\alpha \in [0, 1]$. To this end, we utilize the Neyman-Pearson Lemma [41], which states that for a given $\alpha$, the most powerful rejection rule is threshold-based, i.e., if the likelihood ratio $\frac{P(Y=k)}{P(X=k)}$ is larger than/equal to/smaller than a threshold $h$, $H_0$ is rejected with probability $1/\gamma/0$. More specifically, since $X$ and $Y$ may have different ranges, we divide the discussion into two cases (i.e., Case (1) and Case (2) in Lemma 2). The Neyman-Pearson Lemma [41] is given as follows.

**Lemma 3.** *(Neyman-Pearson Lemma [41]) Let $P$ and $Q$ be probability distributions on $\Omega$ with densities $p$ and $q$, respectively. For the hypothesis testing problem $H_0 : P$ vs $H_1 : Q$, a test $\phi : \Omega \to [0, 1]$ is the most powerful test at level $\alpha$ if and only if there are two constants $h \in [0, +\infty]$ and $\gamma \in [0, 1]$ such that $\phi$ has the form*

$$
\phi(x) = \begin{cases} 1, \text{ if } \frac{q(x)}{p(x)} > h, \\ \gamma, \text{ if } \frac{q(x)}{p(x)} = h, \\ 0, \text{ if } \frac{q(x)}{p(x)} < h, \end{cases} \tag{18}
$$

*and $\mathbb{E}_P[\phi] = \alpha$. The rejection rule suggests that $H_0$ is rejected with a probability of $\phi(x)$ given the observation $x$.*

Given Lemma 3, the problem is then reduced to finding the corresponding $h$ and $\gamma$ such that the type I error rate $\alpha_\phi = \alpha$. For part (1) (the results for part (2) can be shown similarly), we divide the range of $\alpha$ (i.e., $[0, 1]$) into multiple segments, as shown in Fig. 5. To achieve $\alpha = 0$, we set $h = \infty$ and $\gamma = 1$, which suggests that the hypothesis $H_0$ is always rejected when $k < \mathcal{Z}_L^I$ and accepted otherwise. To achieve $\alpha \in (P(X < k), P(X \le k)]$, for $k \in [\mathcal{Z}_L^I, \mathcal{Z}_R^I]$, we set $h = \frac{P(Y=k)}{P(X=k)}$ and $\gamma = \frac{\alpha - P(X < k)}{P(X=k)}$. In this case, it can be shown that $\alpha_\phi = \alpha \in (P(X < k), P(X \le k)]$. To achieve $\alpha \in (P(X < \mathcal{Z}_R^I + 1), 1]$, we set $h = 0$, and $\gamma = \frac{\alpha - P(X < \mathcal{Z}_R^I + 1)}{P(X > \mathcal{Z}_R^I)}$. In this case, it can be shown that $\alpha_\phi = \alpha \in (P(X < \mathcal{Z}_R^I + 1), 1]$. The corresponding $\beta_\phi$ can be derived accordingly, which is given by (16). The complete proof is given below.

*Proof.* Given Lemma 3, the problem is reduced to finding the parameters $h$ and $\gamma$ in (18) such that $\mathbb{E}_P[\phi] = \alpha$, which can be proved as follows.

**Case (1)** We divide $\alpha \in [0, 1]$ into $\mathcal{Z}_R^U - \mathcal{Z}_L^I + 1$ segments: $[P(X < \mathcal{Z}_L^U), P(X < \mathcal{Z}_L^I)] \cup (P(X < \mathcal{Z}_L^I), P(X \le \mathcal{Z}_L^I)] \cup \cdots \cup (P(X < k), P(X \le k)] \cup \cdots \cup (P(X < \mathcal{Z}_R^U), P(X \le \mathcal{Z}_R^U)]$, as shown in Fig. 5.

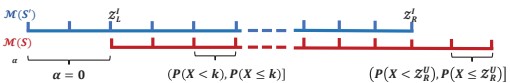

Figure 5: Dividing $\alpha$ into multiple segments for part (1).

When $\alpha = P(X < \mathcal{Z}_L^U) = P(X < \mathcal{Z}_L^I) = 0$, we set $h = +\infty$. In this case, noticing that $\frac{P(Y=k)}{P(X=k)} = h$ for $k < \mathcal{Z}_L^I$, and $\frac{P(Y=k)}{P(X=k)} < h$ otherwise, we have

$$
\mathbb{E}_P[\phi] = \gamma P(X < \mathcal{Z}_L^I) = 0 = \alpha, \tag{19}
$$

and

$$
\beta_\phi^+(0) = 1 - \mathbb{E}_Q[\phi] = 1 - \gamma P(Y < \mathcal{Z}_L^I). \tag{20}
$$

The infimum is attained when $\gamma = 1$, which yields $\beta_\phi^+(0) = P(Y \ge \mathcal{Z}_L^I)$.

When $\alpha \in (P(X < k), P(X \le k)]$ for $k \in [\mathcal{Z}_L^I, \mathcal{Z}_R^I]$, we set $h = \frac{P(Y=k)}{P(X=k)}$. In this case, $\frac{P(Y=k')}{P(X=k')} = h$ for $k' = k$, and $\frac{P(Y=k')}{P(X=k')} > h$ for $k' < k$, and therefore

$$
\mathbb{E}_P[\phi] = P(X < k) + \gamma P(X = k). \tag{21}
$$

We adjust $\gamma$ such that $\mathbb{E}_P[\phi] = \alpha$, which yields

$$
\gamma = \frac{\alpha - P(X < k)}{P(X = k)}, \tag{22}
$$

and

$$
\begin{aligned}
\beta_\phi^+(\alpha) &= 1 - [P(Y < k) + \gamma P(Y = k)] \\
&= P(Y \ge k) - P(Y = k)\frac{\alpha - P(X < k)}{P(X = k)} \\
&= P(Y \ge k) + \frac{P(Y = k)P(X < k)}{P(X = k)} - \frac{P(Y = k)}{P(X = k)}\alpha
\end{aligned} \tag{23}
$$

When $\alpha \in (P(X < k), P(X \leq k)]$ for $k \in (\mathcal{Z}_R^I, \mathcal{Z}_R^U]$, we set $h = 0$. In this case, $\frac{P(Y=k')}{P(X=k')} = h$ for $k' > \mathcal{Z}_R^I$, and $\frac{P(Y=k')}{P(X=k')} > h$ for $k' \leq \mathcal{Z}_R^I$. As a result,

$$\mathbb{E}_P[\phi] = P(X \leq \mathcal{Z}_R^I) + \gamma P(X > \mathcal{Z}_R^I), \tag{24}$$

and

$$\beta_\phi^+(\alpha) = 1 - [P(Y \leq \mathcal{Z}_R^I) + \gamma P(Y > \mathcal{Z}_R^I)] = 0 \tag{25}$$

Similarly, we can prove the second part of Lemma 2 as follows.

**Case (2)** We also divide $\alpha \in [0, 1]$ into $\mathcal{Z}_R^U - \mathcal{Z}_L^I + 1$ segments: $[P(X > \mathcal{Z}_L^U), P(X \geq \mathcal{Z}_L^U)] \cup \cdots \cup (P(X > k), P(X \geq k)] \cup \cdots \cup (P(X > \mathcal{Z}_R^I), P(X \geq \mathcal{Z}_R^I)]$, as shown in Fig. 6.

Figure 6: Dividing $\alpha$ in to multiple segments for part (2).

When $\alpha \in (P(X > k), P(X \geq k)]$ for $k \in [\mathcal{Z}_L^U, \mathcal{Z}_L^I)$, we set $h = 0$. In this case,

$$\mathbb{E}_P[\phi] = P(X \geq \mathcal{Z}_L^I) + \gamma P(X < \mathcal{Z}_L^I), \tag{26}$$

and

$$\beta_\phi^-(\alpha) = 1 - [P(Y \geq \mathcal{Z}_L^I) + \gamma P(Y < \mathcal{Z}_L^I)] = 0 \tag{27}$$

When $\alpha \in (P(X > k), P(X \geq k)]$ for $k \in [\mathcal{Z}_L^I, \mathcal{Z}_R^I]$, we set $h = \frac{P(Y=k)}{P(X=k)}$. In this case,

$$\mathbb{E}_P[\phi] = P(X > k) + \gamma P(X = k). \tag{28}$$

Setting $\mathbb{E}_P[\phi] = \alpha$ yields

$$\gamma = \frac{\alpha - P(X > k)}{P(X = k)}, \tag{29}$$

and

$$
\begin{aligned}
&\beta_\phi^-(\alpha) \\
&= 1 - [P(Y > k) + \gamma P(Y = k)] \\
&= P(Y \leq k) - P(Y = k)\frac{\alpha - P(X > k)}{P(X = k)} \\
&= P(Y \leq k) + \frac{P(Y = k)P(X > k)}{P(X = k)} - \frac{P(Y = k)}{P(X = k)}\alpha
\end{aligned}
\tag{30}
$$

When $\alpha = P(X > \mathcal{Z}_R^I) = 0$, we set $h = +\infty$. In this case,

$$\mathbb{E}_P[\phi] = \gamma P(X > \mathcal{Z}_R^I) = 0 = \alpha, \tag{31}$$

and

$$\beta_\phi^+(0) = 1 - \mathbb{E}_Q[\phi] = 1 - \gamma P(Y > \mathcal{Z}_R^I). \tag{32}$$

The infimum is attained when $\gamma = 1$, which yields $\beta_\phi^-(0) = P(Y \leq \mathcal{Z}_R^I)$. $\square$

# B Proofs of Theoretical Results

## B.1 Proof of Theorem 1

**Theorem 1.** *Let $\tilde{Z} = Binom(M, p)$, the binomial noise mechanism in Algorithm 1 is $f^{bn}(\alpha)$-differentially private with*

$$f^{bn}(\alpha) = \min\{\beta_{\phi,\inf}^+(\alpha), \beta_{\phi,\inf}^-(\alpha)\}, \tag{33}$$

*in which*

$$
\beta_{\phi,\inf}^+(\alpha) = 
\begin{cases}
P(\tilde{Z} \geq \tilde{k} + l) + \frac{P(Z=\tilde{k}+l)P(\tilde{Z}<\tilde{k})}{P(\tilde{Z}=\tilde{k})} - \frac{P(\tilde{Z}=\tilde{k}+l)}{P(\tilde{Z}=\tilde{k})}\alpha, \\
\qquad\qquad\qquad \text{for } \alpha \in [P(\tilde{Z} < \tilde{k}), P(\tilde{Z} \leq \tilde{k})], \tilde{k} \in [0, M - l], \\
0, \qquad\qquad\qquad\qquad\qquad\qquad \text{for } \alpha \in [P(\tilde{Z} \leq M - l), 1].
\end{cases}
\tag{34}
$$

$$\beta_{\phi,\text{inf}}^-(\alpha) = \begin{cases} P(\tilde{Z} \leq \tilde{k} - l) + \frac{P(\tilde{Z}=\tilde{k}-l)P(\tilde{Z}>\tilde{k})}{P(\tilde{Z}=\tilde{k})} - \frac{P(\tilde{Z}=\tilde{k}-l)}{P(\tilde{Z}=\tilde{k})}\alpha, \\ \qquad\qquad \text{for } \alpha \in [P(\tilde{Z} > \tilde{k}), P(\tilde{Z} \geq \tilde{k})], \tilde{k} \in [l, M], \\ 0, \qquad\qquad\qquad\qquad\qquad\qquad \text{for } \alpha \in [P(\tilde{Z} \geq l), 1]. \end{cases} \qquad (35)$$

Given that $P(\tilde{Z} = k) = \binom{M}{k}p^k(1-p)^{M-k}$, it can be readily shown that when $p = 0.5$, both $\beta_{\phi,\text{inf}}^+(\alpha)$ and $\beta_{\phi,\text{inf}}^-(\alpha)$ are maximized, and $f(\alpha) = \beta_{\phi,\text{inf}}^+(\alpha) = \beta_{\phi,\text{inf}}^-(\alpha)$.

Before proving Theorem 1, we first show the following lemma.

**Lemma 4.** *Let $X = x_i + Binom(M, p)$ and $Y = x_i' + Binom(M, p)$. Then, if $x_i > x_i'$,*
$$\beta_\phi^+(\alpha) = \begin{cases} P(Y \geq k) + \frac{P(Y=k)P(X<k)}{P(X=k)} - \frac{P(Y=k)}{P(X=k)}\alpha, \\ \qquad\qquad \text{if } \alpha \in [P(X < k), P(X \leq k)],\ k \in [x_i, x_i' + M]. \\ 0, \qquad\qquad\qquad \text{if } \alpha \in (P(X < x_i' + M + 1), 1]. \end{cases} \qquad (36)$$
*If $x_i < x_i'$,*
$$\beta_\phi^-(\alpha) = \begin{cases} P(Y \leq k) + \frac{P(Y=k)P(X>k)}{P(X=k)} - \frac{P(Y=k)}{P(X=k)}\alpha, \\ \qquad\qquad \text{if } \alpha \in [P(X > k), P(X \geq k)],\ k \in [x_i', x_i + M]. \\ 0, \qquad\qquad\qquad \text{if } \alpha \in (P(X > x_i' - 1), 1] \end{cases} \qquad (37)$$

*Proof of Lemma 4.* When $x_i > x_i'$, it can be easily verified that $P(X = k) > 0$ only for $k \in [x_i, x_i + 1, \cdots, x_i + M]$, $P(Y = k) > 0$ only for $k \in [x_i', x_i' + 1, \cdots, x_i' + M]$. For $k \in [x_i, \cdots, x_i' + M]$, we have
$$\begin{aligned}
\frac{P(Y = k)}{P(X = k)} &= \frac{\binom{M}{k-x_i'}p^{k-x_i'}(1-p)^{M-k+x_i'}}{\binom{M}{k-x_i}p^{k-x_i}(1-p)^{M-k+x_i}} \\
&= \frac{(N-k+x_i'+1)(N-k+x_i'+2)\cdots(N-k+x_i)}{(k-x_i+1)(k-x_i+2)\cdots(k-x_i')}(\frac{1-p}{p})^{x_i'-x_i}.
\end{aligned} \qquad (38)$$
It can be observed that $\frac{P(Y=k)}{P(X=k)}$ is a decreasing function of $k$.

When $x_i < x_i'$, it can be easily verified that $P(X = k) > 0$ only for $k \in [x_i, x_i + 1, \cdots, x_i + M]$, $P(Y = k) > 0$ only for $k \in [x_i', x_i' + 1, \cdots, x_i' + M]$. For $k \in [x_i', \cdots, x_i + M]$, we have
$$\begin{aligned}
\frac{P(Y = k)}{P(X = k)} &= \frac{\binom{M}{k-x_i'}p^{k-x_i'}(1-p)^{M-k+x_i'}}{\binom{M}{k-x_i}p^{k-x_i}(1-p)^{M-k+x_i}} \\
&= \frac{(k-x_i'+1)(k-x_i'+2)\cdots(k-x_i)}{(N-k+x_i+1)(N-k+x_i+2)\cdots(N-k+x_i')}(\frac{1-p}{p})^{x_i'-x_i}.
\end{aligned} \qquad (39)$$
It can be observed that $\frac{P(Y=k)}{P(X=k)}$ is an increasing function of $k$, and invoking Lemma 2 completes the proof. $\qquad\square$

Given Lemma 4, we are ready to prove Theorem 1.

*Proof of Theorem 1.* Let $\tilde{Z} = Binom(M, p)$, $X = x_i + \tilde{Z}$ and $Y = x_i' + \tilde{Z}$. Two cases are considered:

**Case 1:** $x_i > x_i'$.

In this case, according to Lemma 4, we have
$$\beta_\phi^+(\alpha) = \begin{cases} P(Y \geq k) + \frac{P(Y=k)P(X<k)}{P(X=k)} - \frac{P(Y=k)}{P(X=k)}\alpha, \\ \qquad\qquad \text{for } \alpha \in [P(X < k), P(X \leq k)],\ k \in [x_i, x_i' + M], \\ 0, \qquad\qquad\qquad \text{for } \alpha \in [P(X \leq x_i' + M), 1], \end{cases} \qquad (40)$$

In the following, we show the infimum of $\beta_\phi^+(\alpha)$. For the ease of presentation, let $\tilde{k} = k - x_i$ and $x_i - x_i' = \Delta$. Then, we have

$$
\begin{aligned}
P(Y \geq k) &= P(x_i' + \tilde{Z} \geq k) = P(\tilde{Z} \geq \tilde{k} + \Delta), \\
P(Y = k) &= P(\tilde{Z} = \tilde{k} + \Delta), \\
P(X < k) &= P(x_i + \tilde{Z} < k) = P(\tilde{Z} < \tilde{k}), \\
P(X = k) &= P(x_i + \tilde{Z} = k) = P(\tilde{Z} = \tilde{k}).
\end{aligned}
\tag{41}
$$

(40) can be rewritten as

$$
\beta_\phi^+(\alpha) = \begin{cases} P(\tilde{Z} \geq \tilde{k} + \Delta) + \frac{P(\tilde{Z}=\tilde{k}+\Delta)P(\tilde{Z}<\tilde{k})}{P(\tilde{Z}=\tilde{k})} - \frac{P(\tilde{Z}=\tilde{k}+\Delta)}{P(\tilde{Z}=\tilde{k})}\alpha, & \text{for } \alpha \in [P(\tilde{Z} < \tilde{k}), P(\tilde{Z} \leq \tilde{k})], \\ & \tilde{k} \in [0, M - \Delta], \\ 0, & \text{for } \alpha \in [P(Z \leq M - \Delta), 1]. \end{cases}
\tag{42}
$$

Let $J(\Delta, \tilde{k}) = P(\tilde{Z} \geq \tilde{k} + \Delta) + \frac{P(\tilde{Z}=\tilde{k}+\Delta)P(\tilde{Z}<\tilde{k})}{P(\tilde{Z}=\tilde{k})} - \frac{P(\tilde{Z}=\tilde{k}+\Delta)}{P(\tilde{Z}=\tilde{k})}\alpha$, we have

$$
\begin{aligned}
J(\Delta + 1, \tilde{k}) - J(\Delta, \tilde{k}) = &-P(\tilde{Z} = \tilde{k} + \Delta) \\
&+ \frac{P(\tilde{Z} = \tilde{k} + \Delta + 1) - P(\tilde{Z} = \tilde{k} + \Delta)}{P(\tilde{Z} = \tilde{k})}[P(\tilde{Z} < \tilde{k}) - \alpha].
\end{aligned}
\tag{43}
$$

Since $\alpha \in [P(\tilde{Z} < \tilde{k}), P(\tilde{Z} \leq \tilde{k})]$, we have $P(\tilde{Z} < \tilde{k}) - \alpha \in [-P(\tilde{Z} = \tilde{k}), 0]$. If $P(\tilde{Z} = \tilde{k} + \Delta + 1) - P(\tilde{Z} = \tilde{k} + \Delta) > 0$, $J(\Delta + 1, \tilde{k}) - J(\Delta, \tilde{k}) < -P(\tilde{Z} = \tilde{k} + \Delta) < 0$. If $P(\tilde{Z} = \tilde{k} + \Delta + 1) - P(\tilde{Z} = \tilde{k} + \Delta) < 0$, $J(\Delta + 1, \tilde{k}) - J(\Delta, \tilde{k}) < -P(\tilde{Z} = \tilde{k} + \Delta + 1) < 0$. As a result, the infimum of $\beta_\phi^+(\alpha)$ is attained when $\Delta = l$, i.e., $x_i = l$ and $x_i' = 0$, which yields

$$
\beta_{\phi,\text{inf}}^+(\alpha) = \begin{cases} P(\tilde{Z} \geq \tilde{k} + l) + \frac{P(\tilde{Z}=\tilde{k}+l)P(\tilde{Z}<\tilde{k})}{P(Z=\tilde{k})} - \frac{P(\tilde{Z}=\tilde{k}+l)}{P(\tilde{Z}=\tilde{k})}\alpha, \\ \qquad \text{for } \alpha \in [P(\tilde{Z} < \tilde{k}), P(\tilde{Z} \leq \tilde{k})], \tilde{k} \in [0, M - l], \\ 0, \qquad \text{for } \alpha \in [P(\tilde{Z} \leq M - l), 1]. \end{cases}
\tag{44}
$$

**Case 2:** $x_i < x_i'$.

In this case, according to Lemma 4, we have

$$
\beta_\phi^-(\alpha) = \begin{cases} P(Y \leq k) + \frac{P(Y=k)P(X>k)}{P(X=k)} - \frac{P(Y=k)}{P(X=k)}\alpha, \\ \qquad \text{for } \alpha \in [P(X > k), P(X \geq k)], k \in [x_i', x_i + M], \\ 0, \qquad \text{for } \alpha \in [P(X \geq x_i'), 1], \end{cases}
\tag{45}
$$

In the following, we show the infimum of $\beta(\alpha)$. For the ease of presentation, let $\tilde{k} = k - x_i$ and $x_i' - x_i = \Delta$. Then, we have

$$
\begin{aligned}
P(Y \leq k) &= P(x_i' + \tilde{Z} \leq k) = P(\tilde{Z} \leq \tilde{k} - \Delta), \\
P(Y = k) &= P(\tilde{Z} = \tilde{k} - \Delta), \\
P(X > k) &= P(x_i + \tilde{Z} > k) = P(\tilde{Z} > \tilde{k}), \\
P(X = k) &= P(x_i + \tilde{Z} = k) = P(\tilde{Z} = \tilde{k}).
\end{aligned}
\tag{46}
$$

(45) can be rewritten as

$$
\beta_\phi^-(\alpha) = \begin{cases} P(\tilde{Z} \leq \tilde{k} - \Delta) + \frac{P(\tilde{Z}=\tilde{k}-\Delta)P(\tilde{Z}>\tilde{k})}{P(\tilde{Z}=\tilde{k})} - \frac{P(\tilde{Z}=\tilde{k}-\Delta)}{P(\tilde{Z}=\tilde{k})}\alpha, \\ \qquad \text{for } \alpha \in [P(\tilde{Z} > \tilde{k}), P(\tilde{Z} \geq \tilde{k})], \tilde{k} \in [\Delta, M], \\ 0, \qquad \text{for } \alpha \in [P(\tilde{Z} \geq \Delta), 1]. \end{cases}
\tag{47}
$$

Let $J(\Delta, \tilde{k}) = P(\tilde{Z} \leq \tilde{k} - \Delta) + \frac{P(\tilde{Z}=\tilde{k}-\Delta)P(\tilde{Z}>\tilde{k})}{P(\tilde{Z}=\tilde{k})} - \frac{P(\tilde{Z}=\tilde{k}-\Delta)}{P(\tilde{Z}=\tilde{k})}\alpha$, we have

$$
\begin{aligned}
J(\Delta + 1, \tilde{k}) - J(\Delta, \tilde{k}) = &-P(\tilde{Z} = \tilde{k} - \Delta) \\
&+ \frac{P(\tilde{Z} = \tilde{k} - \Delta - 1) - P(\tilde{Z} = \tilde{k} - \Delta)}{P(\tilde{Z} = \tilde{k})}[P(\tilde{Z} > \tilde{k}) - \alpha]
\end{aligned}
\tag{48}
$$

Since $\alpha \in [P(\tilde{Z} > \tilde{k}), P(\tilde{Z} \geq \tilde{k})]$, we have $P(\tilde{Z} > \tilde{k}) - \alpha \in [-P(\tilde{Z} = \tilde{k}), 0]$. If $P(\tilde{Z} = \tilde{k} - \Delta - 1) - P(\tilde{Z} = \tilde{k} - \Delta) > 0$, then $J(\Delta + 1, \tilde{k}) - J(\Delta, \tilde{k}) < -P(\tilde{Z} = \tilde{k} - \Delta) < 0$. If $P(\tilde{Z} = \tilde{k} - \Delta - 1) - P(\tilde{Z} = \tilde{k} - \Delta) < 0$, then $J(\Delta + 1, \tilde{k}) - J(\Delta, \tilde{k}) < -P(\tilde{Z} = \tilde{k} - \Delta - 1) < 0$. As a result, the infimum of $\beta_\phi^-(\alpha)$ is attained when $\Delta = l$, i.e., $x_i = 0$ and $x_i' = l$, which yields

$$\beta_{\phi,\inf}^-(\alpha) = \begin{cases} P(\tilde{Z} \leq \tilde{k} - l) + \frac{P(\tilde{Z} = \tilde{k} - l)P(\tilde{Z} > \tilde{k})}{P(\tilde{Z} = \tilde{k})} - \frac{P(\tilde{Z} = \tilde{k} - l)}{P(\tilde{Z} = \tilde{k})}\alpha, \\ \qquad\qquad \text{for } \alpha \in [P(\tilde{Z} > \tilde{k}), P(\tilde{Z} \geq \tilde{k})], \tilde{k} \in [l, M], \\ 0, \qquad\qquad\qquad\qquad\qquad\qquad \text{for } \alpha \in [P(\tilde{Z} \geq l), 1]. \end{cases} \qquad (49)$$

Combining (44) and (49) completes the first part of the proof. When $p = 0.5$, it can be found that both $\beta_{\phi,\inf}^+(\alpha)$ and $\beta_{\phi,\inf}^-(\alpha)$ are maximized, and $f(\alpha) = \beta_{\phi,\inf}^+(\alpha) = \beta_{\phi,\inf}^-(\alpha)$. $\qquad\square$

## B.2 Proof of Theorem 2

**Theorem 2.** *The binomial mechanism in Algorithm 2 is $f^{bm}(\alpha)$-differentially private with*
$$f^{bm}(\alpha) = \min\{\beta_{\phi,\inf}^+(\alpha), \beta_{\phi,\inf}^-(\alpha)\}, \qquad (50)$$

*in which*
$$\beta_{\phi,\inf}^+(\alpha) = 1 - [P(Y < k) + \gamma P(Y = k)] = P(Y \geq k) + \frac{P(Y = k)P(X < k)}{P(X = k)} - \frac{P(Y = k)}{P(X = k)}\alpha,$$
*for $\alpha \in [P(X < k), P(X \leq k)]$ and $k \in \{0, 1, 2, \cdots, M\}$, where $X = Binom(M, p_{max})$ and $Y = Binom(M, p_{min})$, and*
$$\beta_{\phi,\inf}^-(\alpha) = 1 - [P(Y > k) + \gamma P(Y = k)] = P(Y \leq k) + \frac{P(Y = k)P(X > k)}{P(X = k)} - \frac{P(Y = k)}{P(X = k)}\alpha,$$
*for $\alpha \in [P(X > k), P(X \geq k)]$ and $k \in \{0, 1, 2, \cdots, M\}$, where $X = Binom(M, p_{min})$ and $Y = Binom(M, p_{max})$. When $p_{max} = 1 - p_{min}$, we have $\beta_{\phi,\inf}^+(\alpha) = \beta_{\phi,\inf}^-(\alpha)$.*

*Proof.* Observing that the output space of the binomial mechanism remains the same for different data $x_i$, i.e., $\mathcal{Z}_L^I = \mathcal{Z}_L^U = 0$ and $\mathcal{Z}_R^I = \mathcal{Z}_R^U = M$ in Lemma 2. Moreover, let $X = Binom(M, p)$ and $Y = Binom(M, q)$, we have $\frac{P(Y=k)}{P(X=k)} = \frac{\binom{M}{k}q^k(1-q)^{M-k}}{\binom{M}{k}p^k(1-p)^{M-k}} = (\frac{1-q}{1-p})^M(\frac{q(1-p)}{p(1-q)})^k$. Similarly, we consider the following two cases.

**Case 1:** $q < p$.

In this case, we can find that $\frac{P(Y=k)}{P(X=k)}$ is a decreasing function of $k$. Therefore, according to Lemma 2, we have

$$\begin{aligned} \beta_\phi^+(\alpha) &= 1 - [P(Y < k) + \gamma P(Y = k)] \\ &= P(Y \geq k) - P(Y = k)\frac{\alpha - P(X < k)}{P(X = k)} \\ &= P(Y \geq k) + \frac{P(Y = k)P(X < k)}{P(X = k)} - \frac{P(Y = k)}{P(X = k)}\alpha \end{aligned} \qquad (51)$$

In the following, we show that the infimum is attained when $p = p_{max}$ and $q = p_{min}$. For Binomial distribution $Y$, we have $\frac{\partial P(Y<k)}{\partial q} \leq 0$ and $\frac{\partial P(Y\leq k)}{\partial q} \leq 0, \forall k$.

$$\begin{aligned} \frac{\partial \beta_\phi^+(\alpha)}{\partial q} &= -\frac{\partial P(Y < k)}{\partial q} - \gamma \frac{\partial P(Y = k)}{\partial q} \\ &= -(1 - \gamma)\frac{\partial P(Y < k)}{\partial q} - \gamma \frac{\partial P(Y \leq k)}{\partial q} \\ &\geq 0. \end{aligned} \qquad (52)$$

Therefore, the infimum is attained when $q = p_{min}$.

Suppose $X = Binom(M, p)$ and $\hat{X} = Binom(M, \hat{p})$. Without loss of generality, assume $p > \hat{p}$. Suppose that $\alpha \in [P(X < k), P(X \leq k)]$ and $\alpha \in [P(\hat{X} < \hat{k}), P(\hat{X} \leq \hat{k})]$ for some $k$ and $\hat{k}$ are satisfied simultaneously, it can be readily shown that $k \geq \hat{k}$. In addition, $\alpha \in [\max\{P(X <$

$k), P(\hat{X} < \hat{k})\}, \min\{P(X \leq k), P(\hat{X} \leq \hat{k})\}]$. Let

$$\beta_{\phi,p}^+(\alpha) = P(Y \geq k) + \frac{P(Y = k)[P(X < k) - \alpha]}{P(X = k)}, \tag{53}$$

and

$$\beta_{\phi,\hat{p}}^+(\alpha) = P(Y \geq \hat{k}) + \frac{P(Y = \hat{k})[P(\hat{X} < \hat{k}) - \alpha]}{P(\hat{X} = \hat{k})}, \tag{54}$$

$$\begin{aligned}
&\beta_{\phi,p}^+(\alpha) - \beta_{\phi,\hat{p}}^+(\alpha) \\
&= P(Y \geq k) - P(Y \geq \hat{k}) + \frac{P(Y = k)[P(X < k) - \alpha]}{P(X = k)} - \frac{P(Y = \hat{k})[P(\hat{X} < \hat{k}) - \alpha]}{P(\hat{X} = \hat{k})} \\
&= P(Y > k) - P(Y > \hat{k}) + \frac{P(Y = k)[P(X \leq k) - \alpha]}{P(X = k)} - \frac{P(Y = \hat{k})[P(\hat{X} \leq \hat{k}) - \alpha]}{P(\hat{X} = \hat{k})}.
\end{aligned} \tag{55}$$

Obviously, $P(Y \geq k) - P(Y \geq \hat{k}) \leq 0$ and $P(Y > k) - P(Y > \hat{k}) \leq 0$ for $k \geq \hat{k}$. Observing that $\beta_{\phi,p}^+(\alpha) - \beta_{\phi,\hat{p}}^+(\alpha)$ is a linear function of $\alpha \in [\max\{P(X < k), P(\hat{X} < \hat{k})\}, \min\{P(X \leq k), P(\hat{X} \leq \hat{k})\}]$ given $Y, X, \hat{X}, k$ and $\hat{k}$, we consider the following four possible cases:

**1)** $P(X < k) \leq P(\hat{X} < \hat{k})$ **and** $\alpha = P(\hat{X} < \hat{k})$**:** In this case, $\frac{P(Y=k)[P(X<k)-\alpha]}{P(X=k)} = \frac{P(Y=k)[P(X<k)-P(\hat{X}<\hat{k})]}{P(X=k)} \leq 0$. As a result, $\beta_{\phi,p}^+(\alpha) - \beta_{\phi,\hat{p}}^+(\alpha) \leq 0$.

**2)** $P(X < k) > P(\hat{X} < \hat{k})$ **and** $\alpha = P(X < k)$**:** In this case,

$$\begin{aligned}
&\beta_{\phi,p}^+(\alpha) - \beta_{\phi,\hat{p}}^+(\alpha) \\
&= P(Y \geq k) - P(Y \geq \hat{k}) + \frac{P(Y = k)[P(X < k) - \alpha]}{P(X = k)} - \frac{P(Y = \hat{k})[P(\hat{X} < \hat{k}) - \alpha]}{P(\hat{X} = \hat{k})} \\
&= P(Y \geq k) - P(Y \geq \hat{k}) - \frac{P(Y = \hat{k})[P(\hat{X} < \hat{k}) - P(X < k)]}{P(\hat{X} = \hat{k})}.
\end{aligned} \tag{56}$$

When $k = \hat{k}$, since $p > \hat{p}$, we have $P(\hat{X} < \hat{k}) - P(X < \hat{k}) > 0$, which violates the condition that $P(X < k) > P(\hat{X} < \hat{k})$.

When $k > \hat{k}$, we have $P(Y \geq k) - P(Y \geq \hat{k}) \leq -P(Y = \hat{k})$. Therefore,

$$\begin{aligned}
\beta_{\phi,p}^+(\alpha) - \beta_{\phi,\hat{p}}^+(\alpha) &\leq -P(Y = \hat{k}) - \frac{P(Y = \hat{k})[P(\hat{X} < \hat{k}) - P(X < k)]}{P(\hat{X} = \hat{k})} \\
&= -\frac{P(Y = \hat{k})[P(\hat{X} \leq \hat{k}) - P(X < k)]}{P(\hat{X} = \hat{k})} \\
&\leq 0.
\end{aligned} \tag{57}$$

**3)** $P(X \leq k) \leq P(\hat{X} \leq \hat{k})$ **and** $\alpha = P(X \leq k)$**:** In this case,

$$\begin{aligned}
&\frac{P(Y = k)[P(X \leq k) - \alpha]}{P(X = k)} - \frac{P(Y = \hat{k})[P(\hat{X} \leq \hat{k}) - \alpha]}{P(\hat{X} = \hat{k})} = \\
&\quad -\frac{P(Y = \hat{k})[P(\hat{X} \leq \hat{k}) - P(X \leq k)]}{P(\hat{X} = \hat{k})} \leq 0
\end{aligned} \tag{58}$$

As a result, $\beta_{\phi,p}^+(\alpha) - \beta_{\phi,\hat{p}}^+(\alpha) \leq P(Y > k) - P(Y > \hat{k}) \leq 0$.

**4)** $P(X \leq k) > P(\hat{X} \leq \hat{k})$ **and** $\alpha = P(\hat{X} \leq \hat{k})$**:** In this case, when $k = \hat{k}$, $P(X \leq k) - P(\hat{X} \leq \hat{k}) > 0$, which violates the condition that $P(X \leq k) > P(\hat{X} \leq \hat{k})$.

When $k > \hat{k}$,

$$\beta^+_{\phi,p}(\alpha) - \beta^+_{\phi,\hat{p}}(\alpha)$$

$$= P(Y \geq k) - P(Y \geq \hat{k}) + \frac{P(Y = k)[P(X < k) - P(\hat{X} \leq \hat{k})]}{P(X = k)}$$

$$- \frac{P(Y = \hat{k})[P(\hat{X} < \hat{k}) - P(\hat{X} \leq \hat{k})]}{P(\hat{X} = \hat{k})} \tag{59}$$

$$= P(Y \geq k) - P(Y > \hat{k}) + \frac{P(Y = k)[P(X < k) - P(\hat{X} \leq \hat{k})]}{P(X = k)}.$$

Since $k > \hat{k}$, $P(Y \geq k) - P(Y > \hat{k}) \leq 0$. In addition, $P(X < k) - P(\hat{X} \leq \hat{k}) \leq 0$ since $\alpha \in [\max\{P(X < k), P(\hat{X} < \hat{k})\}, P(\hat{X} \leq \hat{k})]$. As a result, $\beta^+_{\phi,p}(\alpha) - \beta^+_{\phi,\hat{p}}(\alpha) \leq P(Y > k) - P(Y > \hat{k}) \leq 0$.

Now that $\beta^+_{\phi,p}(\alpha) - \beta^+_{\phi,\hat{p}}(\alpha)$ is a linear function of $\alpha \in [\max\{P(X < k), P(\hat{X} < \hat{k})\}, \min\{P(X \leq k), P(\hat{X} \leq \hat{k})\}]$, which is non-positive in the extreme points (i.e., the boundaries), we can conclude that $\beta^+_{\phi,p}(\alpha) - \beta^+_{\phi,\hat{p}}(\alpha) \leq 0$ for any $\alpha \in [\max\{P(X < k), P(\hat{X} < \hat{k})\}, \min\{P(X \leq k), P(\hat{X} \leq \hat{k})\}]$. Therefore, the infimum of $\beta^+_\phi(\alpha)$ is attained when $p = p_{max}$.

**Case 2:** $q > p$.

In this case, we can find that $\frac{P(Y=k)}{P(X=k)}$ is an increasing function of $k$. As a result, according to Lemma 2, we have

$$\beta^-_\phi(\alpha) = P(Y \leq k) + \frac{P(Y = k)P(X > k)}{P(X = k)} - \frac{P(Y = k)}{P(X = k)}\alpha. \tag{60}$$

Similarly, it can be shown that the infimum is attained when $q = p_{max}$ and $p = p_{min}$.

As a result, we have

$$T(P, Q)(\alpha) = \min\{\beta^+_{\phi,\inf}(\alpha), \beta^-_{\phi,\inf}(\alpha)\} \tag{61}$$

$\square$

### B.3 Proof of Theorem 3

**Theorem 3.** *The ternary stochastic compressor is* $f^{ternary}(\alpha)$*-differentially private with*

$$f^{ternary}(\alpha) = \begin{cases} 1 - \frac{A+c}{A-c}\alpha, & \text{for } \alpha \in [0, \frac{A-c}{2B}], \\ 1 - \frac{c}{B} - \alpha, & \text{for } \alpha \in [\frac{A-c}{2B}, 1 - \frac{A+c}{2B}], \\ \frac{A-c}{A+c} - \frac{A-c}{A+c}\alpha, & \text{for } \alpha \in [1 - \frac{A+c}{2B}, 1]. \end{cases} \tag{62}$$

We provide the $f$-DP analysis for a generic ternary stochastic compressor defined as follows.

**Definition 8 (Generic Ternary Stochastic Compressor).** *For any given* $x \in [-c, c]$*, the generic compressor* $ternary$ *outputs* $ternary(x, p_1, p_0, p_{-1})$*, which is given by*

$$ternary(x, p_1, p_0, p_{-1}) = \begin{cases} 1, & \text{with probability } p_1(x), \\ 0, & \text{with probability } p_0, \\ -1, & \text{with probability } p_{-1}(x), \end{cases} \tag{63}$$

*where* $p_0$ *is the design parameter that controls the level of sparsity and* $p_1(x), p_{-1}(x) \in [p_{min}, p_{max}]$*. It can be readily verified that* $p_1 = \frac{A+x}{2B}, p_0 = 1 - \frac{A}{B}, p_{-1} = \frac{A-x}{2B}$ *(and therefore* $p_{min} = \frac{A-c}{2B}$ *and* $p_{max} = \frac{A+c}{2B}$*) for the ternary stochastic compressor in Definition 6.*

In the following, we show the $f$-DP of the generic ternary stochastic compressor, and the corresponding $f$-DP guarantee for the compressor in Definition 6 can be obtained with $p_{min} = \frac{A-c}{2B}$, $p_{max} = \frac{A+c}{2B}$, and $p_0 = 1 - \frac{A}{B}$.

**Lemma 5.** *Suppose that $p_0$ is independent of $x$, $p_{max} + p_{min} = 1 - p_0$, and $p_1(x) > p_1(y), \forall x > y$. The ternary compressor is $f^{ternary}(\alpha)$-differentially private with*

$$f^{ternary}(\alpha) = \begin{cases} 1 - \frac{p_{max}}{p_{min}}\alpha, & \text{for } \alpha \in [0, p_{min}], \\ p_0 + 2p_{min} - \alpha, & \text{for } \alpha \in [p_{min}, 1 - p_{max}], \\ \frac{p_{min}}{p_{max}} - \frac{p_{min}}{p_{max}}\alpha, & \text{for } \alpha \in [1 - p_{max}, 1], \end{cases} \tag{64}$$

*Proof.* Similar to the binomial mechanism, the output space of the ternary mechanism remains the same for different inputs. Let $Y = ternary(x_i', p_1, p_0, p_{-1})$ and $X = ternary(x_i, p_1, p_0, p_{-1})$, we have

$$\frac{P(Y = -1)}{P(X = -1)} = \frac{p_{-1}(x_i')}{p_{-1}(x_i)},$$

$$\frac{P(Y = 0)}{P(X = 0)} = 1, \tag{65}$$

$$\frac{P(Y = 1)}{P(X = 1)} = \frac{p_1(x_i')}{p_1(x_i)}.$$

When $x_i > x_i'$, it can be observed that $\frac{P(Y=k)}{P(X=k)}$ is a decreasing function of $k$. According to Lemma 2, we have

$$\beta_\phi^+(\alpha) = \begin{cases} 1 - \frac{p_{-1}(x_i')}{p_{-1}(x_i)}\alpha, & \text{for } \alpha \in [0, p_{-1}(x_i)], \\ p_0 + p_1(x_i') + p_{-1}(x_i) - \alpha, & \text{for } \alpha \in [p_{-1}(x_i), 1 - p_1(x_i)], \\ \frac{p_1(x_i')}{p_1(x_i)} - \frac{p_1(x_i')}{p_1(x_i)}\alpha, & \text{for } \alpha \in [1 - p_1(x_i), 1]. \end{cases} \tag{66}$$

When $x_i < x_i'$, it can be observed that $\frac{P(Y=k)}{P(X=k)}$ is an increasing function of $k$. According to Lemma 2, we have

$$\beta_\phi^-(\alpha) = \begin{cases} 1 - \frac{p_1(x_i')}{p_1(x_i)}\alpha, & \text{for } \alpha \in [0, p_1(x_i)], \\ p_0 + p_{-1}(x_i') + p_1(x_i) - \alpha, & \text{for } \alpha \in [p_1(x_i), 1 - p_{-1}(x_i)], \\ \frac{p_{-1}(x_i')}{p_{-1}(x_i)} - \frac{p_{-1}(x_i')}{p_{-1}(x_i)}\alpha, & \text{for } \alpha \in [1 - p_{-1}(x_i), 1]. \end{cases} \tag{67}$$

The infimum of $\beta_\phi^+(\alpha)$ is attained when $p_{-1}(x_i') = p_{max}$ and $p_{-1}(x_i) = p_{min}$, while the infimum of $\beta_\phi^-(\alpha)$ is attained when $p_1(x_i') = p_{max}$ and $p_1(x_i) = p_{min}$. As a result, we have

$$f^{ternary}(\alpha) = \begin{cases} 1 - \frac{p_{max}}{p_{min}}\alpha, & \text{for } \alpha \in [0, p_{min}], \\ p_0 + 2p_{min} - \alpha, & \text{for } \alpha \in [p_{min}, 1 - p_{max}], \\ \frac{p_{min}}{p_{max}} - \frac{p_{min}}{p_{max}}\alpha, & \text{for } \alpha \in [1 - p_{max}, 1], \end{cases} \tag{68}$$

which completes the proof. $\qquad\square$

### B.4 Proof of Theorem 4

**Theorem 4.** *Given a vector $x_i = [x_{i,1}, x_{i,2}, \cdots, x_{i,d}]$ with $|x_{i,j}| \le c, \forall j$. Applying the ternary compressor to the $j$-th coordinate of $x_i$ independently yields $\mu$-GDP with $\mu = -2\Phi^{-1}(\frac{1}{1+(\frac{A+c}{A-c})^d})$.*

Before proving Theorem 4, we first introduce the following lemma.

**Lemma 6.** *[42, 43] Any $(\epsilon, 0)$-DP algorithm is also $\mu$-GDP for $\mu = -2\Phi^{-1}(\frac{1}{1+e^\epsilon})$, in which $\Phi(\cdot)$ is the cumulative density function of normal distribution.*

*Proof.* According to Theorem 3, in the scalar case, the ternary stochastic compressor is $f^{ternary}(\alpha)$-differentially private with

$$f^{ternary}(\alpha) = \begin{cases} 1 - \frac{A+c}{A-c}\alpha, & \text{for } \alpha \in [0, \frac{A-c}{2B}], \\ 1 - \frac{c}{B} - \alpha, & \text{for } \alpha \in [\frac{A-c}{2B}, 1 - \frac{A+c}{2B}], \\ \frac{A-c}{A+c} - \frac{A-c}{A+c}\alpha, & \text{for } \alpha \in [1 - \frac{A+c}{2B}, 1]. \end{cases} \tag{69}$$

It can be easily verified that $f^{ternary}(\alpha) \geq \max\{0, 1 - (\frac{A+c}{A-c})\alpha, (\frac{A-c}{A+c})(1-\alpha)\}$. Invoking Lemma 1 suggests that it is $(\log(\frac{A+c}{A-c}), 0)$-DP. Extending it to the $d$-dimensional case yields $(d\log(\frac{A+c}{A-c})^M, 0)$-DP. As a result, according to Lemma 6, it is $-2\Phi^{-1}(\frac{1}{1+(\frac{A+c}{A-c})^d})$-GDP. $\qquad\square$

## B.5 Proof of Theorem 5

**Theorem 5.** *For a vector $x_i = [x_{i,1}, x_{i,2}, \cdots, x_{i,d}]$ with $|x_{i,j}| \leq c, \forall j$, the ternary compressor with $B \geq A > c$ is $f^{ternary}(\alpha)$-DP with*

$$G_\mu(\alpha + \gamma) - \gamma \leq f^{ternary}(\alpha) \leq G_\mu(\alpha - \gamma) + \gamma, \tag{70}$$

*in which*

$$\mu = \frac{2\sqrt{d}c}{\sqrt{AB - c^2}}, \quad \gamma = \frac{0.56\left[\frac{A-c}{2B}\left|1 + \frac{c}{B}\right|^3 + \frac{A+c}{2B}\left|1 - \frac{c}{B}\right|^3 + \left(1 - \frac{A}{B}\right)\left|\frac{c}{B}\right|^3\right]}{(\frac{A}{B} - \frac{c^2}{B^2})^{3/2}d^{1/2}}. \tag{71}$$

Before proving Theorem 5, we first define the following functions as in [15],

$$\mathrm{kl}(f) = -\int_0^1 \log|f'(x)|dx, \tag{72}$$

$$\kappa_2(f) = \int_0^1 \log^2|f'(x)|dx, \tag{73}$$

$$\kappa_3(f) = \int_0^1 |\log|f'(x)||^3 dx, \tag{74}$$

$$\bar{\kappa}_3(f) = \int_0^1 |\log|f'(x)| + \mathrm{kl}(f)|^3 dx. \tag{75}$$

The central limit theorem for $f$-DP is formally introduced as follows.

**Lemma 7** ([15]). *Let $f_1, ..., f_n$ be symmetric trade-off functions such that $\kappa_3(f_i) < \infty$ for all $1 \leq i \leq d$. Denote*

$$\mu = \frac{2\|kl\|_1}{\sqrt{\|\kappa_2\|_1 - \|kl\|_2^2}}, \text{and} \quad \gamma = \frac{0.56\|\bar{\kappa}_3\|_1}{(\|\kappa_2\|_1 - \|kl\|_2^2)^{3/2}},$$

*and assume $\gamma < \frac{1}{2}$. Then, for all $\alpha \in [\gamma, 1 - \gamma]$, we have*

$$G_\mu(\alpha + \gamma) - \gamma \leq f_1 \otimes f_2 \otimes \cdots \otimes f_d(\alpha) \leq G_\mu(\alpha - \gamma) + \gamma. \tag{76}$$

Given Lemma 7, we are ready to prove Theorem 5.

*Proof.* Given $f_i(\alpha)$ in (62), we have

$$
\begin{aligned}
\mathrm{kl}(f) &= -\left[\frac{A-c}{2B}\log\left(\frac{A+c}{A-c}\right) + \frac{A+c}{2B}\log\left(\frac{A-c}{A+c}\right)\right] \\
&= \left[\frac{A+c}{2B} - \frac{A-c}{2B}\right]\log\left(\frac{A+c}{A-c}\right) \\
&= \frac{c}{B}\log\left(\frac{A+c}{A-c}\right),
\end{aligned} \tag{77}
$$

$$
\begin{aligned}
\kappa_2(f) &= \left[\frac{A-c}{2B}\log^2\left(\frac{A+c}{A-c}\right) + \frac{A+c}{2B}\log^2\left(\frac{A-c}{A+c}\right)\right] \\
&= \frac{A}{B}\log^2\left(\frac{A+c}{A-c}\right),
\end{aligned} \tag{78}
$$

$$
\begin{aligned}
\kappa_3(f) &= \left[\frac{A-c}{2B}\left|\log\left(\frac{A+c}{A-c}\right)\right|^3 + \frac{A+c}{2B}\left|\log\left(\frac{A-c}{A+c}\right)\right|^3\right] \\
&= \frac{A}{B}\left|\log\left(\frac{A+c}{A-c}\right)\right|^3,
\end{aligned} \tag{79}
$$

$$\bar{\kappa}_3(f) = \left[\frac{A-c}{2B}\left|1 + \frac{c}{B}\right|^3 + \frac{A+c}{2B}\left|1 - \frac{c}{B}\right|^3 + \left(1 - \frac{A}{B}\right)\left|\frac{c}{B}\right|^3\right]\left|\log\left(\frac{A+c}{A-c}\right)\right|^3. \tag{80}$$

The corresponding $\mu$ and $\gamma$ are given as follows

$$\mu = \frac{2d\frac{c}{B}}{\sqrt{\frac{A}{B}d - \frac{c^2}{B^2}d}} = \frac{2\sqrt{d}c}{\sqrt{AB - c^2}}, \tag{81}$$

$$\gamma = \frac{0.56 \left[\frac{A-c}{2B}\left|1 + \frac{c}{B}\right|^3 + \frac{A+c}{2B}\left|1 - \frac{c}{B}\right|^3 + \left(1 - \frac{A}{B}\right)\left|\frac{c}{B}\right|^3\right]}{(\frac{A}{B} - \frac{c^2}{B^2})^{3/2}d^{1/2}}, \tag{82}$$

which completes the proof. □

## C $f$-DP of the Poisson Binomial Mechanism

The Poisson binomial mechanism [9] is presented in Algorithm 3. In the following, we show the

---

**Algorithm 3** Poisson Binomial Mechanism

---

**Input**: $p_i \in [p_{min}, p_{max}], \forall i \in \mathcal{N}$

Privatization: $Z_{pb} \triangleq PB(p_1, p_2, \cdots, p_N) = \sum_{i \in \mathcal{N}} Binom(M, p_i)$.

---

$f$-DP guarantee of the Poisson binomial mechanism with $M = 1$. The extension to the proof for $M > 1$ is straightforward by following a similar technique.

**Theorem 6.** *The Poisson binomial mechanism with $M = 1$ in Algorithm 3 is $f^{pb}(\alpha)$-differentially private with*

$$f^{pb}(\alpha) = \min\left\{\max\left\{0, 1 - \frac{1 - p_{min}}{1 - p_{max}}\alpha, \frac{p_{min}}{p_{max}}(1 - \alpha)\right\},\right.$$
$$\left.\max\left\{0, 1 - \frac{p_{max}}{p_{min}}\alpha, \frac{1 - p_{max}}{1 - p_{min}}(1 - \alpha)\right\}\right\}. \tag{83}$$

*Proof.* For Poisson Binomial, let
$$X = PB(p_1, p_2, \cdots, p_{i-1}, p_i, p_{i+1}, \cdots, p_N),$$
$$Y = PB(p_1, p_2, \cdots, p_{i-1}, p'_i, p_{i+1}, \cdots, p_N), \tag{84}$$
$$Z = PB(p_1, p_2, \cdots, p_{i-1}, p_{i+1}, \cdots, p_N),$$
in which $PB$ stands for Poisson Binomial. In this case,
$$\frac{P(Y = k+1)}{P(X = k+1)} = \frac{P(Z = k+1)(1 - p'_i) + P(Z = k)p'_i}{P(Z = k+1)(1 - p_i) + P(Z = k)p_i}. \tag{85}$$

In addition,
$$P(Y = k+1)P(X = k) - P(Y = k)P(X = k+1)$$
$$= [P(Z = k+1)P(Z = k-1) - (P(Z = k))^2](p_i - p'_i). \tag{86}$$
Since $P(Z = k+1)P(Z = k-1) - (P(Z = k))^2 < 0$ for Poisson Binomial distribution, we have
$$P(Y = k+1)P(X = k) - P(Y = k)P(X = k+1)\begin{cases} > 0, & \text{if } p_i < p'_i, \\ < 0, & \text{if } p_i > p'_i. \end{cases} \tag{87}$$

That being said, $\frac{P(Y=k)}{P(X=k)}$ is an increasing function of $k$ if $p_i < p'_i$ and a decreasing function of $k$ if $p_i > p'_i$. Following the same analysis as that in the proof of Theorem 2, for $p_i > p'_i$, we have
$$\beta_\phi^+(\alpha) = 1 - [P(Y < k) + \gamma P(Y = k)]$$
$$= P(Y \geq k) - P(Y = k)\frac{\alpha - P(X < k)}{P(X = k)} \tag{88}$$
$$= P(Y \geq k) + \frac{P(Y = k)P(X < k)}{P(X = k)} - \frac{P(Y = k)}{P(X = k)}\alpha,$$
for $\alpha \in [P(X < k), P(X \leq k)]$ and $k \in \{0, 1, 2, \cdots, N\}$.

In the following, we show that the infimum of $\beta_\phi^+(\alpha)$ is attained when $p_i = p_{max}$ and $p'_i = p_{min}$.

**Case 1:** $k = 0$. In this case,

$$P(Y \geq 0) = 1,$$
$$P(Y = 0) = P(Z = 0)(1 - p_i'),$$
$$P(X < 0) = 0,$$
$$P(X = 0) = P(Z = 0)(1 - p_i).$$

(89)

Plugging (89) into (88) yields

$$\beta_\phi^+(\alpha) = 1 - \frac{1 - p_i'}{1 - p_i}\alpha.$$

(90)

It is obvious that the infimum is attained when $p_i = p_{max}$ and $p_i' = p_{min}$.

**Case 2:** $k > 0$. In this case,

$$P(Y \geq k) = P(Z \geq k) + P(Z = k - 1)p_i',$$
$$P(Y = k) = P(Z = k)(1 - p_i') + P(Z = k - 1)p_i',$$
$$P(X < k) = P(Z < k) - P(Z = k - 1)p_i,$$
$$P(X = k) = P(Z = k)(1 - p_i) + P(Z = k - 1)p_i.$$

(91)

Plugging (91) into (88) yields

$$\beta_\phi^+(\alpha) = p(Z > k) + P(Z = k)p_i' + [P(X \leq k) - \alpha]\frac{[P(Z = k) - [P(Z = k) - P(Z = k - 1)p_i']]}{P(X = k)}.$$

(92)

The $p_i'$ related term is given by

$$\left[\frac{P(X = k)P(Z = k)}{P(X = k)} - \frac{[P(Z = k) - P(Z = k - 1)][P(X \leq k) - \alpha]}{P(X = k)}\right]p_i'.$$

(93)

Observing that (93) is a linear function of $\alpha$, we only need to examine $\alpha \in \{P(X < k), P(X \leq k)\}$. More specifically, when $\alpha = P(X \leq k)$, it is reduced to $P(Z = k)p_i'$; when $\alpha = P(X < k)$, it is reduced to $P(Z = k - 1)p_i'$. In both cases, the infimum is attained when $p_i' = p_{min}$.

Given that $p_i' = p_{min}$, the same technique as in the proof of Theorem 2 can be applied to show that the infimum is attained when $p = p_{max}$.

Since $\frac{P(Y=k)}{P(X=k)}$ is a decreasing function of $k$ when $p_i > p_i'$, we have

$$\frac{p_{min}}{p_{max}} \leq \frac{P(Y = k)}{P(X = k)} \leq \frac{1 - p_{min}}{1 - p_{max}}.$$

(94)

Given that $\beta_\phi^+(\alpha)$ is a decreasing function of $\alpha$ with $\beta_\phi^+(0) = 1$ and $\beta_\phi^+(1) = 0$, we can readily conclude that $\beta_\phi^+(\alpha) \geq \max\{0, 1 - \frac{1-p_{min}}{1-p_{max}}\alpha\}$ and $\beta_\phi^+(\alpha) \geq \frac{p_{min}}{p_{max}}(1 - \alpha)$. That being said, $\beta_\phi^+(\alpha) \geq \max\{0, 1 - \frac{1-p_{min}}{1-p_{max}}\alpha, \frac{p_{min}}{p_{max}}(1 - \alpha)\}$.

Similarly, for $p_i < p_i'$, we have

$$\beta_\phi^-(\alpha) = 1 - [P(Y > k) + \gamma P(Y = k)]$$

$$= P(Y \leq k) - P(Y = k)\frac{\alpha - P(X > k)}{P(X = k)}$$

(95)

$$= P(Y \leq k) + \frac{P(Y = k)P(X > k)}{P(X = k)} - \frac{P(Y = k)}{P(X = k)}\alpha$$

for $\alpha \in [P(X > k), P(X \geq k)]$ and $k \in \{0, 1, 2, \cdots, N\}$. The infimum is attained when $p_i = p_{min}$, $p_i' = p_{max}$.

Since $\frac{P(Y=k)}{P(X=k)}$ is an increasing function of $k$ when $p_i < p_i'$, we have

$$\frac{1 - p_{max}}{1 - p_{min}} \leq \frac{P(Y = k)}{P(X = k)} \leq \frac{p_{max}}{p_{min}}.$$

(96)

Given that $\beta_\phi^-(\alpha)$ is an increasing function of $\alpha$ with $\beta_\phi^-(0) = 1$ and $\beta_\phi^-(1) = 0$, we can easily conclude that $\beta_\phi^-(\alpha) \geq \max\{0, 1 - \frac{p_{max}}{p_{min}}\alpha\}$ and $\beta_\phi^-(\alpha) \geq \frac{1-p_{max}}{1-p_{min}}(1 - \alpha)$. That being said, $\beta_\phi^-(\alpha) \geq \max\{0, 1 - \frac{p_{max}}{p_{min}}\alpha, \frac{1-p_{max}}{1-p_{min}}(1 - \alpha)\}$. $\qquad\square$

