# OpenReview forum: "Breaking the Communication-Privacy-Accuracy Tradeoff with $f$-Differential Privacy"
_NeurIPS.cc/2023/Conference — NeurIPS 2023 poster_

### Official Review · Reviewer_mVNo · 2023-06-22

**Soundness:** 3 good
**Presentation:** 3 good
**Contribution:** 1 poor
**Rating:** 3
**Confidence:** 3

**Summary:**

This paper studies distributed mean estimation under privacy and communication constraints. This paper focuses on characterizing the recently defined notion of $f$-DP for communication-efficient mechanisms, where $f$-DP can be converted to the standard $(\epsilon,\delta)$-DP. The paper analyzed the $f$-DP of different known mechanisms in the literature. Furthermore, a new algorithm named ternary compression is proposed.

**Strengths:**

The distributed mean estimation is an important topic and has lots of applications in federated learning under privacy and communication constraints. The presentation is well written.

**Weaknesses:**

Although the topic is interesting, my main concerns are about the contributions of this paper.

 * The first part of the paper is to analyze the $f$-DP of well-known discrete mechanisms. While analyzing the $f$-DP is useful, it is not hard to compute it. I don't know what are the major challenges in this analysis and if this analysis provides any new insights into analyzing the $f$-DP. Hence, in my opinion, this part isn't novel enough as a main contribution of this paper.
* The expressions of the $f$-DP are not in a closed form; the expressions are functions of a set of probabilities that might be computationally expensive to compute for large system parameters.
* The proposed mechanism of ternary compression is not novel. There are lots of similar schemes that don't send some coordinates with some probability from sampling the coordinates for example [13] and [23]. Actually, the Ternary mechanism can be seen as a combination of CLDP for a set of coordinates that are chosen i.i.d. with probability $B/A$.
* In general, this ternary mechanism has higher privacy because it has lower accuracy, since with some non-zero probability (when $B>A$), the client will not send anything.

*  The optimal trade-offs between privacy, communication, and accuracy have already been characterized for LDP in different works in the literature, and hence, I don't understand what is the major difference in this paper in comparison with the existing work.

* Typos: please define the abbreviation of GDP (Gaussian differential privacy) for using it. In Algorithm 1, $x_i\in\lbrace 0,1,\ldots,l \rbrace$ instead of square brackets. Figure 4 is small and the lines are close enough so that it is hard to distinguish the different mechanisms.

**Questions:**

* It is mentioned in the abstract and also in the introduction that *it remains an open problem whether such discrete-valued mechanisms provide any privacy protection.*
I don't know which open problem the authors referring to. If it is the distributed mean estimation under jointly local differential privacy and communication constraints, then this problem is already solved in the literature.

* In line 51, *SQKR doesn't account for the privacy introduced during sparsification.* I don't understand this part. AFAIK, SQKR is order optimal, and hence, it doesn't lose privacy analysis.

* In lines 259-261, what do these numbers of $\epsilon$ refer to, and from where it comes?

* In lines 331-333, *we essentially remove the dependency of accuracy on the communication overhead.* Could you please explain more? In general, it is supposed that there is a trade-off between communication and accuracy, so the accuracy depends on the communication budget.

**Limitations:**

Yes

---

> ### Author Rebuttal · Authors · 2023-08-09
>
> Dear Reviewer mVNo,
>
> We appreciate your time in reviewing our paper and providing helpful comments. We believe that your concerns are due to misunderstandings. Different from existing methods (e.g., SQKR) which ignore the privacy amplification in compression, the proposed ternary compressor achieves much better privacy guarantees. For example, as we discussed in our global response, the ternary compressor improves the privacy guarantees for SQKR from $\epsilon_{SQKR} = \\{1,2,5\\}$ to $\epsilon_{ternary} = \\{0.05,0.2,3.9\\}$ **given the same communication cost and MSE**. Please find our detailed response below.
>
> **Comment 1**: The main challenge of the analyses lies in the fact that the tradeoff functions are piece-wise functions with both the domain and the range of each piece determined by both the mechanism and the datasets, and finding the infimums analytically is highly non-trivial. Generally, since the piece-wise tradeoff functions are mechanism-dependent, the analyses are also mechanism specific. We adopt completely different strategies in finding the infimums for the binomial noise and the Poisson binomial mechanism.
>
> Our analyses advance the literature by deriving tighter privacy guarantees for the binomial noise and deriving local DP guarantees for the binomial mechanism in $f$-DP, which is of vital importance in their practical use. More importantly, based on our analyses, we propose the ternary compressor that utilizes compression for privacy amplification (to the best of our knowledge, this is the first work that achieves it), which outperforms the existing methods and delivers almost identical privacy-accuracy performance to the classic Gaussian mechanism while improving communication efficiency. This is of significant importance in applications like federated learning when communication and privacy are the major bottlenecks.
>
> **Comment 2:** The probabilities are not expensive to compute. All the random variables in Theorem 1 and Theorem 2 follow a binomial distribution with known parameters (i.e., $M$ and $p$). Both their pdfs and cdfs are known analytically and easy to compute numerically. Therefore, we consider the corresponding $f$-DP guarantees are in closed-form expressions. In fact, the tradeoff functions are piecewise linear functions with fixed slopes for each piece. We only need to compute the boundary points for each piece, i.e., at most $O(M)$ cdfs values of the binomial distribution are needed to obtain Eq. (5) and Eq. (6) (similarly for Theorem 2), which is computationally affordable. We believe the concern is due to different interpretations of "closed-form" and will revise carefully.
>
> **Comment 3, 4, 5, Question 2:** There are indeed several schemes that simultaneously consider compression and DP. The major difference is that they fail to account for the privacy amplification by sparsification, by exploiting which the proposed ternary mechanism provides significantly better privacy guarantees. For CLDP in [23], as we discussed in Example 2, it is a special case of sto-sign, and the proposed ternary compressor obtains an amplification in privacy (see Fig. 3 in the manuscript). For SQKR in [13], given a $d$-dimensional data $x$, SQKR with $\epsilon$-LDP first samples $k$ out of $d$ coordinates (denote the output by $y$) and applies the $\epsilon$-LDP randomized response mechanism to $y$ (denote the output by $z$). It accounts for the privacy in the process of generating $z$ from $y$, but ignores the privacy in the process of generating $y$ from $x$ (i.e., sparsification). Therefore, these existing methods are only order-optimal (rather than optimal) in characterizing the tradeoff between privacy, communication, and accuracy.
>
> Compared to existing methods, the proposed ternary compressor does not obtain higher privacy by sacrificing accuracy. Instead, we exploit privacy amplification brought by compression (which reduces communication costs) to improve the privacy-accuracy tradeoff. As we discussed in our global response, our experimental results imply that the proposed ternary compressor significantly outperforms SQKR in [13]. We also compare the ternary compressor with the Gaussian mechanism given the same privacy and MSE in the right figure of Fig. 4. Despite the improvement in communication efficiency (at least 32x), the tradeoff between privacy and accuracy for the proposed ternary mechanism matches that of the Gaussian mechanism.
>
> **Question 1:** By open problem, we refer to quantifying the DP guarantees of the compression mechanisms.
>
> **Question 3:** The numbers of $\epsilon$'s refer to the privacy levels in $(\epsilon,\delta)$-DP. In Theorem 3, we derived the $f$-DP guarantee, and the corresponding $\epsilon$ and $\delta$ are obtained by invoking Lemma 1 (line 153).
>
> **Question 4:** Indeed, accuracy depends on the communication budget, and our argument holds under fixed privacy requirements. Since the ternary compressor accounts for privacy amplification in sparsification, the privacy guarantee $\mu$ is closely related to the communication overhead, i.e., the loss in accuracy introduced by communicating less is translated to enhancement in privacy (please see our global response for a detailed discussion). There are two sources of privacy in the ternary compressor: 1) privacy introduced by randomly mapping each coordinate to +1 or -1, and 2) privacy amplification by sparsification. More aggressive sparsification (i.e., a larger $B$) leads to a larger MSE while bringing a larger privacy amplification. Meanwhile, for a target privacy $\mu$, a larger privacy amplification allows us to introduce less randomness (i.e., a smaller $A$) in the random mapping for each coordinate, which results in a smaller MSE. Overall, given the same privacy $\mu$, the MSE remains the same regardless of communication overhead if $\mu < \sqrt{4dr/(1-r)}$. For the existing methods like SQKR, reducing communication leads to degradation in MSE without enhancing privacy.

---

> > ### Comment · Reviewer_mVNo · 2023-08-14
> >
> > Thanks for your rebuttal. Unfortunately, I am still seeing the contributions are limited. The proposed mechanism is not novel as mentioned earlier it can be represented by coordinate sampling and quantization. Due to these reasons, I keep my score.

---

> > > ### Author Response · Authors · 2023-08-15
> > > **Response to the follow-up comment**
> > >
> > > Dear Reviewer mVNo,
> > >
> > > Thanks for your follow-up comment. We cannot agree that the proposed ternary compressor being a combination of coordinate sampling and quantization makes the paper not novel. It is not rare that scientific research builds on prior works. For example, SQKR (reference [13] in the manuscript) combines Kashin's representation [R1], coordinate sampling, and the Randomized Response mechanism (which can be traced back to [R2]), while CLDP (reference [23] in the manuscript) combines the 1-bit quantizer (a special case of the randomized response mechanism) and coordinate sampling. These works make solid contributions to the area by demonstrating satisfactory accuracy/privacy performance or providing rigorous theoretical analyses (or both). Similarly, our contribution and novelty are not only proposing the ternary compressor (in fact, we have already mentioned in lines 243-244 of our manuscript that the ternary compressor is a combination of sign-based quantization and sparsification) but also providing the corresponding theoretical analyses on privacy guarantees in terms of the emerging and promising $f$-DP for both the proposed scheme and the existing mechanisms. For the proposed ternary compressor, its improvement in privacy compared to existing methods is backed by rigorous theoretical analyses. We also advance the literature by providing tighter privacy guarantees for the binomial noise and complementing privacy analysis for the Poisson binomial mechanism by showing its local differential privacy guarantee. Our results reveal that the binomial mechanism captures numerous existing compression and differential privacy mechanisms as special cases, which are valuable on their own.
> > >
> > > More importantly, although there are some existing works (e.g., SQKR) that utilize coordinate sampling and quantization, they adopt coordinate sampling merely for the purpose of improving communication efficiency and fail to account for its impact on privacy from the theoretical aspect. Studying the impact of sparsification on privacy is crucial, especially for applications like differentially private distributed learning where communication costs and privacy are the major bottlenecks and sparsification is one of the most effective and commonly adopted approaches to alleviate the communication burden. To the best of our knowledge, this is the first work that investigates the privacy protection brought by coordinate sampling, which shows that the loss in accuracy due to sparsification can be translated to amplification in privacy and therefore leads to significant improvement in terms of differential privacy guarantees compared to existing methods. Therefore, we believe that the results and findings in this paper are novel and of critical importance to the community.
> > >
> > > Again, we appreciate your time and effort in reading the paper and providing comments. Please let us know if you have further questions.
> > >
> > > Best regards,
> > >
> > > Authors of the paper
> > >
> > > [R1] Y. Lyubarskii and R. Vershynin, “Uncertainty principles and vector quantization,” IEEE Transactions on Information Theory, vol. 56, no. 7, pp. 3491–3501, 2010.
> > >
> > > [R2] S. L. Warner. "Randomized response: A survey technique for eliminating evasive answer bias." Journal of the American Statistical Association, 60(309):63–69, 1965.

---

### Official Review · Reviewer_VYNX · 2023-07-07

**Soundness:** 3 good
**Presentation:** 3 good
**Contribution:** 3 good
**Rating:** 7
**Confidence:** 2

**Summary:**

This paper investigates the f-DP guarantee of several discrete-valued mechanisms in the local-DP model. In particular, closed-form expressions for binomial noise mechanism and Binomial mechanics are derived.

Then, the paper considers the popular problem of aggregating d-dimensional vectors from local users subject to privacy communication constraints. Under the f-DP framework, this paper presents a new "ternary" mechanism and analyzes its f-DP privacy guarantee. Roughly, the ternary mechanism allows each local user to sparsify their vectors by keeping each coordinate with only a small probability and adding noises to the alive coordinates. The hope is that by sparsification and randomization, one can optimize communication (i.e., minimizing the messages from each user to the server) and privacy cost (e.g., each user has its signal "somewhat hidden" in the noises).

The paper's main claim is that, by working with f-DP and the ternary mechanism, we can benefit in **both** privacy and communication by the sparsification process in the ternary mechanism. Intuitively, if the sparsifying threshold is high (i.e., we only keep very few coordinates), we can hope to add less noise to achieve privacy, as the sparsification has already introduced some uncertainty in the data. The paper claims that this is something that prior works cannot offer. The reviewer, unfortunately, did not have a chance to verify the claim.

**Strengths:**

* Working with f-DP **and** in the local model seems like a new approach.
* The proposed algorithm is natural and simple to implement, it is backed by the theory and performs favorably in the experiments.

**Weaknesses:**

* It's a bit hard to interpret/digest the closed-form f-DP guarantee of all these mechanisms. Have some plot/converting to Renyi-DP or (eps,delta)-DP might help.

**Questions:**

* Your algorithm sends **in expection** O(d * A/B) bits. Is there a way to improve this to a worst-case guarantee? How about, say, randomly selecting (A/B * d) coordinates? Is it possible to work out the f-DP (or any other reasonable DP notion) property of this variant?

**Limitations:**

Some future directions are mentioned in the paper.

---

> ### Author Rebuttal · Authors · 2023-08-09
>
> Dear Reviewer VYNX,
>
> We appreciate your time and effort in reviewing and providing a positive evaluation of our work. Please find the point-by-point response to the comments below.
>
> **Comment**: It's a bit hard to interpret/digest the closed-form f-DP guarantee of all these mechanisms. Have some plot/converting to Renyi-DP or $(\epsilon,\delta)$-DP might help.
>
> **Response:** Thanks for the valuable suggestion. We agree that the closed-form expressions characterizing the tradeoff between the two types of error rates are not easy to digest for readers not familiar with $f$-DP, and will add some figures converting it to RDP or $(\epsilon,\delta)$-DP for better illustration. We would like to mention that $f$-DP can be readily converted to $(\epsilon,\delta)$-DP through Lemma 1 in the manuscript by finding $(\epsilon,\delta)$'s such that $f(\alpha) \geq \max\\{0,1-\delta-e^{\epsilon}\alpha,e^{-\epsilon}(1-\delta-\alpha)\\}$. For the privacy guarantees of the binomial noise in Fig. 1, we convert it to $(\epsilon,\delta)$-DP in Remark 1. For the privacy guarantees of the proposed ternary compressor in Fig. 3, we convert it to  $(\epsilon,\delta)$-DP in Remark 3. In our comparison with SQKR in Fig.4 of the manuscript, as we discussed in our global response, given the same MSE and communication cost as that of SQKR with $\epsilon_{SQKR} = \\{1,2,5\\}$, if we translate the privacy guarantees of the ternary compressor from $f$-DP to $\epsilon$-DP via Lemma 1, the proposed ternary compressor yields $\epsilon_{ternary} = \\{0.05,0.2,3.9\\}$. We will add the corresponding discussion in the revised manuscript.
>
> **Question:** Your algorithm sends in expection O(d * A/B) bits. Is there a way to improve this to a worst-case guarantee? How about, say, randomly selecting (A/B * d) coordinates? Is it possible to work out the f-DP (or any other reasonable DP notion) property of this variant?
>
> **Response:** Thanks for the valuable comment. We agree that exploring the worst-case guarantee is crucial. We would like to mention that in the proof of the privacy guarantees, we first consider the scalar case and then extend the results to the multi-dimensional case by invoking the composition theorem to account for the composition. In this case, each coordinate is processed independently. Randomly selecting a fixed number of coordinates may ruin the independence across coordinates and makes the composition more complicated.
>
> That being said, it is also possible that we directly analyze the privacy guarantees in the vector case by invoking Lemma 2 (which characterizes the tradeoff between type I and type II error rates for generic discrete-valued mechanisms) in Appendix A of the manuscript. However, in this case, the range of the randomized mechanism will grow exponentially as the number of selected coordinates increases, and may finally become computationally infeasible. We deem it an interesting direction for extension and will work on it in our future work.
>
> Another straightforward way to avoid the extreme case where too many coordinates remain non-zero and thus little saving in communication is achieved (if this is the concern in the Reviewer's mind) is by incorporating another mechanism that randomly samples a subset of them (with a fixed number of coordinates) before transmission. The privacy guarantee remains the same thanks to the post-processing property of DP. In this case, we lose the privacy amplification introduced by the second sparsification scheme (since we utilize the post-processing property instead of accounting for the privacy amplification). However, for the proposed ternary compressor, the number of coordinates that remain non-zero follows the binomial distribution with a mean of $d\times A/B$ and a variance of $d \times (A/B) \times (1-A/B)$. For the applications like federated learning, $d$ is the size of gradients, which would be in the order of millions for modern neural networks. The central limit theorem tells us that the probabilities of these extreme cases in which too many coordinates remain non-zero are very low.

---

> > ### Comment · Reviewer_VYNX · 2023-08-18
> > **Thank you**
> >
> > Thank you very much to the authors for answering my questions. I would like to keep my current score.

---

### Official Review · Reviewer_Mzdj · 2023-07-07

**Soundness:** 3 good
**Presentation:** 2 fair
**Contribution:** 2 fair
**Rating:** 4
**Confidence:** 2

**Summary:**

This paper analyses the privacy that is provided by stochastic rounding methods when doing distributed mean estimation with local differential privacy. It finds that they can contribute to the privacy guarantee thus achieving a better tradeoff.

It then "breaks" the privacy communication utility trade-off in the sense of pointing out that for the high privacy regime extra communication won't buy you any more accuracy.

**Strengths:**

Previous naive means of implementing DP aggregation with randomised rounding have failed to take advantage of the privacy provided by stochastic compression. This should be utilised if possible which they attempt to do.

**Weaknesses:**

The presentation fails to show a decent comparison showing hwo much is really gained by taking advantage of this randomness. It isn't clear the gain isn't negligible. There is a graph comparing ternary and binary quantisation but it isn't clear whether the parameters actually correspond to low communication, which would require sufficient sparsity.

The supposed break of the three way trade off is really just the observation that in the high privacy regime there is no gain to lots of communication. This is not what break means, the title is overselling.

What is more this observation is very far from new. The idea that O(epsilon) communication suffices goes back a long way for various problems. This was proven with complete generality in https://arxiv.org/pdf/2102.12099.pdf.

**Questions:**

Fixing two of privacy/communication/utility how much does taking advantage of the randomness in the compression actually help the other one?

Am I missing something with the breaking of the trade-off?

**Limitations:**

They don't really talk about the limitations, indeed the limitations aren't really clear from the presentation.

---

> ### Author Rebuttal · Authors · 2023-08-09
>
> Dear Reviewer Mzdj,
>
> We appreciate your time in reviewing our paper and providing constructive comments. We believe that the concerns are mainly due to misunderstanding, please find our response below.
>
> **Our contribution**: We would like to clarify that, as we discussed in the global response, our contribution is more than analyzing the privacy guarantees of stochastic rounding methods in distributed mean estimation with local DP, but advancing the literature by deriving tight privacy guarantees for various DP and compression mechanisms (we believe showing their $f$-DP guarantees is important on its own since it enjoys better composition property than other variants, see e.g., Fig.6 in [R1]), and proposing ternary compressor that exploits privacy amplification in compression.
>
> [R1] El Ouadrhiri, Ahmed, and Ahmed Abdelhadi. "Differential privacy for deep and federated learning: A survey." IEEE Access, 2022.
>
> **Comment:** It isn't clear whether the gain is negligible and the parameters correspond to low communication.
>
> **Response:** We are afraid that this concern is due to a misunderstanding. A larger gain in privacy always corresponds to a larger improvement in communication. We believe that the graph you mention is Fig. 3 in the manuscript. As we discussed in Remark 3, we set $A=0.25$ and $B=0.5$ to generate the figure, which means that the sparsity ratio is 0.5. In addition, the parameters of $A$ and $B$ in Fig. 3 are selected for the purpose of illustration. For the proposed ternary compressor, more aggressive compression (i.e., lower communication) leads to larger privacy amplification. Specifically, the gray area in Fig. 3 corresponds to the privacy improvement (in which $\alpha \in [(A-c)/2B, 1-(A+c)/2B]$ and $f(\alpha) = 1-c/B-\alpha$ in Eq.(13)). For any $A$, increasing $B$ makes the output sparser, and $f(\alpha) = 1-c/B-\alpha$ approaches $f(\alpha) = 1-\alpha$ (which corresponds to perfect privacy). In this sense, as the communication cost decreases to zero (i.e., the output is always 0), the privacy guarantee improves and approaches perfect privacy.
>
> Moreover, as we discussed in our global response, the gain in privacy is not negligible.
>
> **Comment:** The supposed break of the three way trade off is just the observation that in the high privacy regime there is no gain to lots of communication. The title is overselling.
>
> **Response:** We acknowledge that there is rich literature showing that $O(\epsilon)$ communication is sufficient, and this work is partially inspired by [13] which breaks the communication-privacy-accuracy trilemma by proposing SQKR. However, as we discussed in our global response, the message that this paper delivers is not ``communicating less will not hurt the utility much under privacy constraints``, but instead ``the loss in utility caused by communicating less is translated to enhancement in privacy``. Therefore, the results that we present are different from the existing literature, and we do not think the title is overselling. More specifically, existing methods either apply differentially private mechanisms to the compressed output (e.g., SQKR in [13]) or compress the output of differentially private mechanisms (e.g., [R1] mentioned by the reviewer). These mechanisms do not utilize privacy enhancement by compression (which is the focus of this paper). Particularly, the utility of the methods in [R1] depends on the quality of the pseudorandom generator while the utility of SQKR depends on the communication cost $k$. On the contrary, the proposed ternary compressor utilizes privacy amplification in compression, and the tradeoff is essentially only between privacy and accuracy. Since we further improve the results in [13], we also use the word ``break". If the reviewer has better suggestions on the title, we will consider revising it seriously.
>
> Moreover, despite the condition $\mu < \sqrt{4dr/(1-r)}$, our results are not constrained in the high privacy or low privacy regimes. As $r = A/B$ increases and approaches 1 (i.e., no sparsification), the right-hand side of the inequality goes to infinity.
>
> [R1] Feldman, Vitaly, and Kunal Talwar. "Lossless compression of efficient private local randomizers." In International Conference on Machine Learning, 2021.
>
> **Question:** Fixing two of privacy/communication/utility how much does taking advantage of the randomness in the compression actually help the other one?
>
> **Response:** We compare the proposed ternary compressor with SQKR in the left figure of Fig. 4 in the manuscript. As we discussed in our global response, given the same MSE and communication cost as those of SQKR with $\epsilon_{SQKR} = \\{1,2,5\\}$, the proposed ternary compressor attains privacy guarantees of $\epsilon_{ternary} = \\{0.05,0.2,3.9\\}$ by exploiting the privacy amplification in compression. We also compare the proposed ternary compressor with the Gaussian mechanism given the same privacy and utility requirements in the right figure in Fig. 4. It is shown that despite the improvement in communication efficiency (at least 32x if we use 32 bits to represent a float), the tradeoff between privacy and utility for the ternary mechanism matches that of the Gaussian mechanism (i.e., privacy for free).
>
> **Limitations:** We did not add a section to explicitly discuss the limitations due to space constraints, which we will add in our revised manuscript. As we briefly discussed in our discussion below Fig. 4, the improvement in communication efficiency is obtained for free only in the large $d$ regime. Fortunately, in applications like distributed learning, $d$ corresponds to the model size (usually in the orders of millions for modern neural networks). Moreover, despite that the privacy-accuracy tradeoff of the proposed ternary compressor matches that of the Gaussian mechanism which is order-optimal in $(\epsilon,\delta)$-DP, we do not show its optimality by deriving lower bounds in the $f$-DP regime. We will investigate it in future work.

---

> > ### Author Response · Authors · 2023-08-18
> > **Response to Reviewer Mzdj**
> >
> > Dear Reviewer Mzdj,
> >
> > We noticed that you have raised your score from 3 (reject) to 4 (borderline reject), and we appreciate it. We hope that we have clarified our contributions and addressed your concerns in our response above.
> >
> > One notable difference between the proposed method and the existing literature is that we capture the correlation between differential privacy and compression while the existing methods like SQKR do not account for privacy in compression. More specifically, the existing works have shown that in the high privacy regime, the error introduced by compression will be dominated by the error introduced by privacy constraints, while this work (to the best of our knowledge, this is also the first work) further proves that the former can be translated to enhancement in privacy (and therefore provides a better utility-privacy tradeoff as we verified in both theoretical and numerical results). Moreover, we believe that our analyses on the $f$-DP guarantees of the existing differentially private mechanisms and compression schemes are valuable on their own.
> >
> > Please let us know if you have further concerns or comments, and we will further clarify.
> >
> > Best regards,
> >
> > Authors of the paper

---

> > > ### Comment · Reviewer_Mzdj · 2023-08-21
> > >
> > > This makes some sense but the gains still seem small to me. I am therefore neither willing to recommend an acceptance or express much confidence.

---

> > > > ### Author Response · Authors · 2023-08-21
> > > > **Response to Reviewer Mzdj's follow-up comment**
> > > >
> > > > Dear Reviewer Mzdj,
> > > >
> > > > Thanks for your further comment. It is indeed not easy to interpret the privacy guarantee in terms of $f$-DP, and we would like to further clarify from the following different aspects.
> > > >
> > > > **In terms of type II error rate**: Note that as we mentioned in our global response, for the examined scenario in Fig. 4 of the manuscript, compared to SQKR, the proposed ternary compressor increases the type II error rate (which measures the probability that the attacker makes a wrong guess about the true data in hypothesis testing) from $f(\alpha) = 0.068$ to $f(\alpha) = 0.484$ (when $\alpha=0.5$ and $\epsilon = 2$).
> > > >
> > > > **In terms of $\epsilon$-DP given the same MSE and communication cost**: As we discussed in our global response, given the same MSE and communication cost as that of SQKR with $\epsilon_{SQKR} = \\{1,2,5\\}$, the proposed ternary compressor yields approximately $\epsilon_{ternary} = \\{0.05,0.2,3.9\\}$. Note that by definition, for differentially private mechanisms $M$ and all neighboring datasets $D_{1}$ and $D_2$, $e^{\epsilon} = \max_{S}\frac{P(M(D_{1}) \in S)}{P(M(D_{2}) \in S)}$, in which $S$ is the subset of the image of $M$. In this sense, $e^{\epsilon}$ measures the ratio of probabilities that the attacker observing $S$ given neighboring datasets $D_{1}$ and $D_2$. A smaller ratio implies that it is more difficult for the attacker to distinguish $D_{1}$ and $D_{2}$. As an example, improving from $\epsilon_{SQKR} = 5$ to $\epsilon_{ternary} = 3.9$ means that this ratio of probabilities decreases by a factor of $e^{5-3.9} \approx 3$.
> > > >
> > > > **In terms of MSE given the same $\epsilon$ and communication cost**: Recall that the MSE of SQKR is proportional to $var = \frac{1}{k}(\frac{e^{\epsilon}+2^{k}-1}{e^{\epsilon}-1})^{2}$ (here we only keep the terms that are related to the communication cost and the privacy requirement $\epsilon$ for ease of discussion), and $\epsilon_{SQKR} = 5$ corresponds to $var = 6.31$ (let $k =10$). We note that for the same $var = 6.31$, the ternary compressor yields $\epsilon_{ternary} = 3.9$.
> > > >
> > > > We can also readily examine the MSE of SQKR when it attains $\epsilon_{SQKR} = 3.9$ (i.e., the same privacy as that of the ternary compressor with $var = 6.31$), which gives $var = 49.08$. This means that to achieve the privacy guarantee of $\epsilon_{SQKR} = 3.9$, SQKR yields an MSE that is $49.08/6.31 \approx 6.35$ times that of the proposed ternary compressor.
> > > >
> > > > Overall, we believe that such an improvement is not small.
> > > >
> > > > **In terms of the improvement in communication efficiency given the same privacy and utility**: Since we use different measures of differential privacy from SQKR, directly comparing the ternary compressor with SQKR is difficult in this case. Therefore, we adopt the classic Gaussian mechanism (which is order-optimal in terms of privacy-utility tradeoff) as the baseline in the right figure in Fig. 4 of the manuscript. Our results imply that the privacy-utility tradeoff of the proposed ternary compressor matches that of the classic Gaussian mechanism well, i.e., the communication efficiency is obtained with negligible loss. Note that our results hold as long as $\mu < \sqrt{4dr/(1-r)}$, in which $\mu$ is the privacy guarantee and $r$ the sparsity ratio. Even when $r = 1$, the ternary compressor leads to an improvement of 32x in communication efficiency (if 32 bits are used to represent a float) compared to the Gaussian mechanism. The improvement is supposed to be larger when $r$ decreases.
> > > >
> > > > Finally, we would like to mention that the contributions of this paper are not limited to the proposed ternary compressor. The analyses of tight $f$-DP guarantees for a variety of recently proposed differentially privacy mechanisms and compression schemes are valuable on their own.
> > > >
> > > > We hope the above clarification further alleviates your concern. Please let us know if there are further questions.
> > > >
> > > > Best regards,
> > > >
> > > > Authors of the paper

---

### Official Review · Reviewer_191s · 2023-07-10

**Soundness:** 3 good
**Presentation:** 3 good
**Contribution:** 3 good
**Rating:** 6
**Confidence:** 4

**Summary:**

In a federated setting where a server coordinates the collaborative analysis of multiple users with local data, communication efficiency and data privacy are two major issues of consideration. Classical DP mechanisms such as Laplace or Gaussian mechanisms add noises as real numbers -- at the same time, to save communication cost it is preferred to have discrete noises with limited range (resolution). This paper considers all three aspects: communication efficiency, privacy, and accuracy. Specifically, the authors would like to study privacy guarantee of data compression schemes, which intuitively introduces information loss and thus is good for protecting privacy.

The authors use f-DP, and considers binomial mechanisms. Binomial mechanisms have been considered in [6] and [9], this paper provides tighter bounds. In addition, the authors also consider a compression scheme that optimizes for communication efficiency. This is a combination of a sign based quantization and sparsification.

**Strengths:**

I like the motivation of the paper and the consideration of all three objectives, communication efficiency, privacy and accuracy together. The tighter analysis and mechanisms for communication efficiency is of use in practice.

Although the work builds on prior work the improvement by considering compression for privacy amplification is nice. The paper is generally well written.

**Weaknesses:**

A few minor issues, see below.

**Questions:**

Are there any lower bounds on the tradeoff of the three considerations? Any discussion here would be useful.

Writing: it would be good to summarize somewhere in the paper the best recommendation for practitioners, what parameters to choose etc.

Is there general improvement (amplification of privacy) using compression/sparsification? Can the authors elaborate/discuss this?

---

> ### Author Rebuttal · Authors · 2023-08-09
>
> Dear Reviewer 191s,
>
> We appreciate your time and effort in reviewing and providing a positive evaluation of our work. Please find the point-by-point response to the comments below.
>
> **Comment:** Are there any lower bounds on the tradeoff of the three considerations?
>
> **Response:** We fully agree that deriving the lower bounds is of critical importance to evaluate the optimality of the mechanism. For the lower bounds without privacy constraints, Theorem 5.3 in [R1] shows a lower bound of $\Omega(\frac{C^{2}d}{nb})$ for $b$-bit unbiased compression mechanisms, in which $n$ is the number of users. In our case, the MSE for a single user is $O(ABd)$ and the communication cost is $(\log(d)+1)\frac{A}{B}d$ bits. Let $b = (\log(d)+1)\frac{A}{B}d$, we have an MSE of $O(\frac{A^{2}d^{2}(\log(d)+1)}{b})$. Since $A > c = \frac{C}{\sqrt{d}}$, the MSE for $n$ users is therefore given by $O(\frac{C^{2}d(\log(d)+1)}{nb})$, which implies that the proposed mechanism is order-optimal up to a factor of $\log(d)$. Note that the factor of $\log(d)$ is used to represent the index of coordinates that are non-zero, which can be eliminated by allowing for shared randomness.
>
> For the lower bounds given the privacy constraint, unfortunately, different from $(\epsilon,\delta)$-DP for which numerous existing works have derived the lower bounds in mean estimation, there is no such lower bound for $f$-DP in the existing literature. We believe that the analyses are highly non-trivial and deserve independent work.
>
> However, it can be shown that the proposed ternary compressor is order-optimal in terms of $(\epsilon,\delta)$-DP. Particularly, Corollary 2.13 in [R2] shows that a mechanism is $\mu$-GDP if and only if it is $(\epsilon,\delta(\epsilon))$-DP for any $\epsilon \geq 0$ with $\delta(\epsilon) = \Phi(-\frac{\epsilon}{\mu}+\frac{\mu}{2}) - e^{\epsilon}\Phi(-\frac{\epsilon}{\mu}-\frac{\mu}{2})$. In terms of $(\epsilon,\delta)$-DP, the Gaussian mechanism achieves an MSE of $O(\frac{C^{2}d\log(1/\delta)}{n^{2}\epsilon^{2}})$ for central DP (and there will be a loss of factor $n$ for local DP), which is order-optimal (see, e.g., Theorem 3.1 of [R3]). Note that, in terms of $\mu$-GDP, the proposed ternary compressor induces an MSE of $4dC^{2}/\mu^{2} + C^{2} -||x||_{2}^{2}$
>
> while the Gaussian mechanism has an MSE of $4dC^{2}/\mu^{2}$. Notice that the difference $C^{2}-||x||_{2}^{2}$ is a constant and becomes zero when the distribution of $x$ is supported on $[-c,c]^{d}$ (which is adopted in the derivation of the lower bound in [R3]). Since the Gaussian mechanism is order-optimal, the proposed ternary compressor is also order-optimal.
>
> [R1] Chen, Wei-Ning, Christopher A. Choquette Choo, Peter Kairouz, and Ananda Theertha Suresh. "The fundamental price of secure aggregation in differentially private federated learning." In International Conference on Machine Learning, pp. 3056-3089. PMLR, 2022.
>
> [R2] Dong, Jinshuo, Aaron Roth, and Weijie J. Su. "Gaussian differential privacy." Journal of the Royal Statistical Society Series B: Statistical Methodology 84, no. 1 (2022): 3-37.
>
> [R3] Cai, T. Tony, Yichen Wang, and Linjun Zhang. "The cost of privacy: Optimal rates of convergence for parameter estimation with differential privacy." The Annals of Statistics 49, no. 5 (2021): 2825-2850.
>
> **Comment:** It would be good to summarize somewhere in the paper the best recommendation for practitioners, what parameters to choose etc.
>
> **Response:** Thanks for the constructive comment. There are two parameters (i.e., $A$ and $B$) to choose for our proposed ternary compressor. $AB$ determines the $f$-DP guarantees and the MSE while $A/B$ determines the communication cost (in expectation). Utilizing the closed-form expressions that we derive, the corresponding $A$ and $B$ can be readily obtained for any given privacy/MSE and communication cost specifications. We will add more discussions in the discussion part of Section 6.
>
> **Comment:** Is there general improvement (amplification of privacy) using compression/sparsification?
>
> **Response:** This is a great question that our work endeavors to answer. Intuitively, compression and sparsification introduce noise into the system and less useful information is transmitted, which would lead to privacy improvement. However, in terms of DP, compression itself does not necessarily provide privacy guarantees since DP considers the worst-case scenario. Here is a toy example. Suppose that there is a scalar $x\in [-1,1]$, and the user shares $sign(x)$ with the central server. Then, the central server can distinguish any $x > 0$ from $x < 0$ given $sign(x)$, which corresponds to $\epsilon = \infty$ since $\frac{P(sign(x) = +1|x > 0)}{P(sign(x)=+1|x<0)} = \frac{1}{0}$, i.e., no differential privacy is preserved.
>
> Nonetheless, it is likely that sparsification does bring privacy amplification to DP mechanisms. For example, suppose that random sparsification is adopted and a user shares 0 with the server with a probability of $1-\delta$ regardless of its local data $x$. Then the random sparsification scheme itself is $(0,\delta)$-DP, i.e., perfect privacy (when the central server receives 0) with a violation probability of $\delta$ (when the user shares $x$). If the shared $x$ is itself differentially private, we believe the random sparsification scheme will further enhance it.
>
> However, mathematically measuring the amplification in privacy may be difficult. In fact, this work constitutes the first step towards this goal by analyzing the tradeoff function between type I and type II error rates in $f$-DP (please see Lemma 2 in Appendix A) for a generic discrete-valued mechanism. Unfortunately, quantifying the $f$-DP guarantees without explicit expression for the compression/sparsification scheme is still infeasible, and we conduct analyses on several popular compression and differentially private mechanisms in this work. We will consider extending the analyses to more general cases in our future work.

---

### Author Rebuttal · Authors · 2023-08-09

Dear Chairs and Reviewers,

The authors would like to thank you for your time in handling our paper and providing insightful comments and suggestions. We are happy to know that the reviewers find our study is of practical use (reviewer 191s), our proposed method is natural and simple to implement while backed by both theory and experiments (reviewer VYNX), the privacy in compression that we study should be utilized if possible (reviewer Mzdj), and the paper is generally well-written (reviewers 191s, reviewer mVNo).

In this work, we investigate the amplification in privacy brought by compression. To this end, we derive the expressions of the tradeoff function between the two types of error rates in the hypothesis testing problem for generic discrete-valued mechanisms, based on which we advance the literature by deriving tighter privacy guarantees for binomial noise and analyzing the local differential privacy guarantees for the Poisson binomial mechanism as well as a variety of discrete-valued differentially private and compression mechanisms. Utilizing the proposed analytical results, we further propose a ternary compressor that exploits privacy amplification in sparsification and show that the loss in accuracy introduced by compression can be translated to enhancement in privacy. Our analyses focus on the recently emerging and promising concept of $f$-DP that enjoys a better composition property than $(\epsilon,\delta)$-DP and Renyi DP, which we believe is valuable to the community on its own.

We find that most of the concerns are due to misunderstandings and believe that we have addressed them adequately. In this global response, we would like to address the following two common concerns.

**The difference from existing methods**: In this work, different from existing methods that combine compression and differentially private mechanisms, we endeavor to exploit the privacy amplification brought by compression. Therefore, what we observe and present is not ``communicating less will not hurt the utility much under privacy constraints``, but instead ``the loss in utility caused by communicating less is translated to enhancement in privacy``. More specifically, the state-of-the-art method SQKR in [R1] yields a variance of $\frac{d}{k}(\frac{e^{\epsilon}+2^{k}-1}{e^{\epsilon}-1})^{2}C^{2}-||x||_{2}^{2}$

for the transmitted signal $x$. While for finite $\epsilon$ and $k$, the MSE may be dominated by error introduced by the privacy requirement $\epsilon$, the error introduced by compression is non-zero and further degrades the MSE, i.e., *the variance is still a function of $k$, and decreasing $k$ leads to an increased MSE without affecting the privacy guarantee $\epsilon$*. On the contrary, for the proposed ternary compressor, the privacy guarantee is given by $\mu = \frac{2\sqrt{d}c}{\sqrt{AB-c^{2}}}$ (line 301 in the revised manuscript), the communication cost is determined by $\frac{A}{B}d$, and the MSE is given by $ABd-||x||_{2}^{2}$.

When the communication cost is reduced with more aggressive sparsification (i.e., $B$ increases for a fixed $A$), there is an increase in MSE, but also a corresponding amplification in privacy, i.e., $\mu$ decreases. In this sense, given a fixed target privacy guarantee $\mu_{target}$, we may decrease $A$ when $B$ increases such that $AB$ (and therefore both $\mu$ and the MSE) remains the same. In this sense, *the MSE is solely determined by the privacy $\mu$ since the error introduced by compression is translated into enhancement in privacy.* With some simple algebra, the MSE of the ternary compressor is given by $(4d/\mu^{2}+1)C^{2}-||x||_{2}^{2}$ when $\mu < \sqrt{4dr/(1-r)}$, which remains the same regardless of the communication cost $r$. In this case, the tradeoff is essentially only between privacy and accuracy, and neither increasing nor decreasing the communication cost affects the tradeoff.

**The gain in privacy compared to SQKR**: The gain in privacy by utilizing the privacy amplification brought by compression is significant. The left figure in Fig. 4 compares the privacy guarantees of the proposed ternary compressor with SQKR given the same communication cost (only 10 out of 250 coordinates are transmitted) and MSE. Note that we transform the $\epsilon$-LDP guarantee of SQKR and the $f$-DP guarantee of the ternary compressor to the tradeoff between type I error rate and type II error rate in hypothesis testing for the ease of a direct comparison. Given the same type I error rate, a larger type II error rate means that the adversary is more likely to make a mistake in the hypothesis testing (i.e., better privacy). It can be observed from the left figure in Fig. 4 that the ternary compressor significantly outperforms SQKR. For example, for SQKR with $\epsilon = 2$, given type I error rate $\alpha = 0.5$, the type II error rate of the attacker is around $f(\alpha) = 0.068$. Meanwhile, given the same MSE and communication cost, the proposed ternary compressor attains a type II error rate of $f(\alpha) = 0.484$ when $\alpha = 0.5$. Given the same MSE and communication cost as that of SQKR with $\epsilon_{SQKR} = \\{1,2,5\\}$, if we translate the privacy guarantees of the ternary compressor from $f$-DP to $\epsilon$-DP via Lemma 1 (we numerically test different $\epsilon$'s such that $f(\alpha) \geq \max\\{0,1-\delta-e^{\epsilon}\alpha,e^{-\epsilon}(1-\delta-\alpha)\\}$ holds for $\delta = 0$), they are approximately $\epsilon_{ternary} = \\{0.05,0.2,3.9\\}$ for the proposed ternary compressor (please see the attached file for the figure). Note that in the high privacy regime, the error introduced by privacy requirements dominates for SQKR, and it becomes more important to exploit the privacy amplification by compression. As a result, a larger gap is observed.

[R1] Chen, Wei-Ning, Peter Kairouz, and Ayfer Ozgur. "Breaking the communication-privacy-accuracy trilemma." Advances in Neural Information Processing Systems 33 (2020): 3312-3324.

---

### Decision · Program_Chairs · 2023-09-21

**Decision:**

Accept (poster)

**Comment:**

The reviewers commend the paper's approach to addressing communication, privacy, and accuracy trade-offs using f-differential privacy. Notably, the focus on improved communication efficiency is appreciated. However, concerns arise regarding the clarity of comparative analysis on the tangible gains. The meta-reviewer recommends acceptance as a poster while urging the author(s) to enhance their comparative demonstrations for better impact.